# Learning from Visual Observation via Offline Pretrained State-to-Go Transformer

**Bohan Zhou**[1]   **Ke Li**[2]   **Jiechuan Jiang**[1]   **Zongqing Lu**[1,2†]
[1] School of Computer Science, Peking University
[2] Beijing Academy of Artificial Intelligence

## Abstract

Learning from visual observation (LfVO), aiming at recovering policies from only visual observation data, is promising yet a challenging problem. Existing LfVO approaches either only adopt inefficient online learning schemes or require additional task-specific information like goal states, making them not suited for open-ended tasks. To address these issues, we propose a two-stage framework for learning from visual observation. In the first stage, we introduce and pretrain State-to-Go (STG) Transformer offline to predict and differentiate latent transitions of demonstrations. Subsequently, in the second stage, the STG Transformer provides intrinsic rewards for downstream reinforcement learning tasks where an agent learns merely from intrinsic rewards. Empirical results on Atari and Minecraft show that our proposed method outperforms baselines and in some tasks even achieves performance comparable to the policy learned from environmental rewards. These results shed light on the potential of utilizing video-only data to solve difficult visual reinforcement learning tasks rather than relying on complete offline datasets containing states, actions, and rewards. The project's website and code can be found at https://sites.google.com/view/stgtransformer.

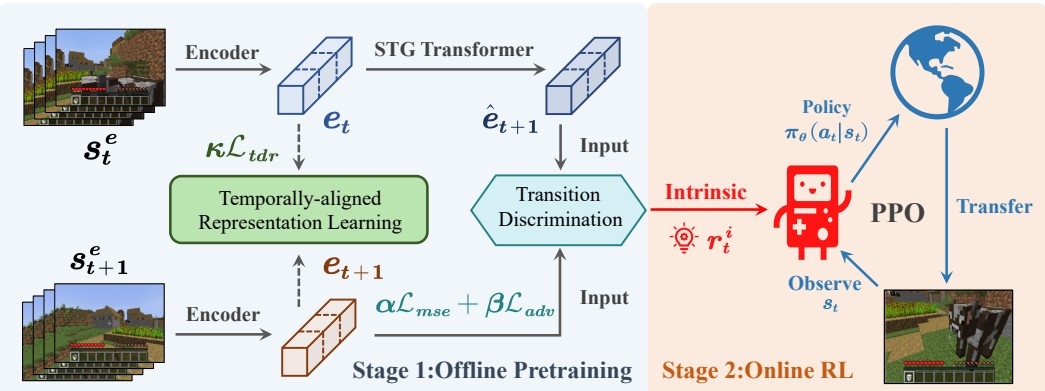

**Figure 1:** A two-stage framework for learning from visual observation. The first stage involves three concurrently pretrained components. A feature encoder is trained in a self-supervised manner to provide easily predicted and temporally aligned representations for stacked-image states. State-to-Go (**STG**) Transformer is trained in an adversarial way to accurately predict transitions in latent space. A discriminator is updated simultaneously to distinguish state transitions of prediction from expert demonstrations, which provides high-quality intrinsic rewards for downstream online reinforcement learning in the next stage.

[†]Corresponding author <zongqing.lu@pku.edu.cn>

37th Conference on Neural Information Processing Systems (NeurIPS 2023).

# 1 Introduction

Reinforcement learning (RL) from scratch imposes significant challenges due to sample inefficiency and hard exploration in environments with sparse rewards. This has led to increased interest in imitation learning (IL). IL agents learn policies by imitating expert demonstrations in a data-driven manner rather than through trial-and-error processes. It has been proven effective in various domains, including games [1] and robotics [2].

However, acquiring demonstrated actions can be expensive or impractical, *e.g.*, from videos that are largely available though, leading to the development of learning from observation (LfO) [3, 4, 5, 6, 7, 8, 9]. This line of research utilizes observation-only data about agent behaviors and state transitions for policy learning. Humans naturally learn from visual observation without requiring explicit action guidance, such as beginners in video games improving their skills by watching skilled players' recordings. However, LfO agents face challenges in extracting useful features from raw visual observations and using them to train a policy due to the lack of explicit action information. Thus, further study of learning from visual observation (LfVO) has the potential to grant agents human-like learning capabilities.

In this paper, we investigate a reinforcement learning setting in which agents learn from visual observation to play challenging video games, such as Atari and Minecraft. Existing LfO approaches like [4, 5, 6, 7, 8] primarily apply to vector-observation environments such as MuJoCo. Others like [3, 9, 10, 11, 12, 13, 14] explicitly consider or can be applied to high-dimensional visual observations. Among them, representation-learning methods [9, 10, 11] learn effective visual representations and recover an intrinsic reward function from them. Adversarial methods [3, 12, 13] learn an expert-agent observation discriminator online to directly indicate visual differences. However, noises or local changes in visual observations may easily cause misclassification [15]. In [12, 13], additional proprioceptive features (*e.g.*, joint angles) are used to train a discriminator, which are unavailable in environments that only provide visual observations. Moreover, as these methods require online training, sample efficiency is much lower compared to offline learning. Goal-oriented methods, like [14], evaluate the proximity of each visual observation to expert demonstrations or predefined goals. However, defining explicit goals is often impractical in open-ended tasks [16]. Furthermore, the continuity of observation sequences in video games cannot be guaranteed due to respawn settings or unanticipated events.

To address these limitations and hence enable RL agents to effectively learn from visual observation, we propose a general two-stage framework that leverages visual observations of expert demonstrations to guide online RL. In the first stage, unlike existing online adversarial methods, we introduce and pretrain **State-to-Go (STG) Transformer**, a variant of Decision Transformer (DT) [17], for **offline** predicting transitions in latent space. In the meanwhile, **temporally-aligned and predictable** visual representations are learned. Together, a discriminator is trained to differentiate expert transitions, generating intrinsic rewards to guide downstream online RL training in the second stage. That said, in the second stage, agents learn merely from generated intrinsic rewards without environmental reward signals. Our empirical evaluation reveals significant improvements in both sample efficiency and overall performance across various video games, demonstrating the effectiveness of the proposed framework.

**Our main contributions are as follows:**

- We propose a general two-stage framework, providing a novel way to enable agents to effectively learn from visual observation. We introduce State-to-Go Transformer, which is pretrained offline merely on visual observations and then employed to guide online reinforcement learning without environmental rewards.

- We simultaneously learn a discriminator and a temporal distance regressor for temporally-aligned embeddings while predicting latent transitions. We demonstrate that the jointly learned representations lead to enhanced performance in downstream RL tasks.

- Through extensive experiments in Atari and Minecraft, we demonstrate that the proposed method substantially outperforms baselines and even achieves performance comparable to the policies learned from environmental rewards in some games, underscoring the potential of leveraging offline video-only data for reinforcement learning.

## 2 Related Work

**Learning from Observation (LfO)** is a more challenging setting than imitation learning (IL), in which an agent learns from a set of demonstrated observations to complete a task without the assistance of action or reward guidance. Many existing works [5, 18, 19, 20] attempt to train an inverse dynamic model (IDM) to label observation-only demonstrations with expert actions, enabling behavior cloning. However, these methods often suffer from compounding error [21]. On the other hand, [4] learns a latent policy and a latent forward model, but the latent actions can sometimes be ambiguous and may not correspond accurately with real actions. GAIfO [3], inspired by [22], is an online adversarial framework that couples the process of learning from expert observations with RL training. GAIfO learns a discriminator to evaluate the similarity between online-collected observations and expert demonstrations. Although helpful in mitigating compounding error [3], it shows limited applicability in environments with high-dimensional observations. Follow-up methods [12, 13] pay more attention to visual-observation environments, but require vector-state in expert observations to either learn a feasible policy or proper visual representations. More importantly, learning a discriminator online is less sample-efficient, compared to LfO via offline pretraining. A recent attempt [23] demonstrates some progress in action-free offline pretraining, but return-to-gos are indispensable in addition to observations because of upside down reinforcement learning (UDRL) framework [24]. Moreover, it only shows satisfactory results in vector-observation environments like MuJoCo. In this work, we focus on learning from offline visual observations (LfVO), a more general and practical setting, which is also considered by the concurrent work VIPER [25]. STG and VIPER both offline pretrain a reward function and then use it for online reinforcement learning. STG learns discriminative intrinsic rewards in WGAN style while VIPER leverages logarithm predictive probability as guidance.

**Visual Representation Learning in RL.** High-quality visual representations are crucial for LfVO. Many previous works [26, 27, 28, 9, 10, 11] have contributed to this in various ways. For example, [26] employs GANs to learn universal representations from different viewpoints, and [27, 28] learn representations via contrastive learning to associate pairs of observations separated by a short time difference. In terms of LfVO, a wide range of methods such as [9, 10, 11] learn temporally continuous representations in a self-supervised manner and utilize temporal distance to assess the progress of demonstrations. Nevertheless, in games like Atari, adjacent image observations may exhibit abrupt or subtle changes due to respawn settings or unanticipated events, not following a gradual change along the timeline. Moreover, over-reliance on temporal information often results in over-optimistic estimates of task progress [14], potentially misleading RL training.

**Transformer in RL.** Transformer [29] is widely acknowledged as a kind of powerful structure for sequence modeling, which has led to domination in a variety of offline RL tasks. Decision Transformer (DT) [17] and Trajectory Transformer (TT) [30] redefine the offline RL problem as a context-conditioned sequential problem to learn an offline policy directly, following the UDRL framework [24]. DT takes states, actions, and return-to-gos as inputs and autoregressively predicts actions to learn a policy. TT predicts the complete sequence dimension by dimension and uses beam search for planning. MGDT [31] samples from a learned return distribution to avoid manually selecting expert-level returns as DT. ODT [32] extends DT to bridge the gap between offline pretraining and online fine-tuning. MADT [33] extends DT to a multi-agent reinforcement learning setting.

## 3 Methodology

### 3.1 Preliminaries

**Reinforcement Learning.** The RL problem can be formulated as a Markov decision process (MDP) [34], which can be represented by a tuple $\mathcal{M} = <\mathcal{S}, \mathcal{A}, \mathcal{P}, \mathcal{R}, \gamma, \rho_0>$. $\mathcal{S}$ denotes the state space and $\mathcal{A}$ denotes the action space. $\mathcal{P} : \mathcal{S} \times \mathcal{A} \times \mathcal{S} \rightarrow [0, 1)$ is the state transition function and $\mathcal{R} : \mathcal{S} \times \mathcal{A} \rightarrow \mathbb{R}$ is the reward function. $\gamma \in [0, 1]$ is the discount factor and $\rho_0 : \mathcal{S} \rightarrow [0, 1]$ represents the initial state distribution. The objective is to find a policy $\pi(a|s) : \mathcal{S} \rightarrow \mathcal{A}$, which maximizes the expected discounted return:

$$J(\pi) = \mathbb{E}_{\rho_0, a_t \sim \pi(\cdot|s_t), s_t \sim \mathcal{P}} \left[ \sum_{t=0}^{\infty} \gamma^t r(s_t, a_t) \right]. \tag{1}$$

**Transformer.** Stacked self-attention layers with residual connections in Transformer is instrumental in processing long-range dependencies, each of which embeds $n$ tokens $\{x_i\}_{i=1}^n$ and outputs $n$ embeddings $\{z_i\}_{i=1}^n$ of the same dimensions considering the information of the whole sequence. In this study, we utilize the GPT [35] architecture, an extension of the Transformer model, that incorporates a causal self-attention mask to facilitate autoregressive generation. Specifically, each input token $x_i$ is mapped to a key $k_i$, a query $q_i$, and a value $v_i$ through linear transformations, where $z_i$ is obtained by computing the weighted sum of history values $v_{1:i}$, with attention weights determined by the normalized dot product between the query $q_i$ and history keys $k_{1:i}$:

$$z_i = \sum_{j=1}^{i} \text{softmax}(\{q_i^\mathsf{T}, k_{j'}\}_{j'=1}^i)_j \cdot v_j. \tag{2}$$

The GPT model only attends to the previous tokens in the sequence during training and inference, thereby avoiding the leakage of future information, which is appropriate in state prediction.

**Learning from Observation.** The goal is to learn a policy from an expert state sequence dataset $\mathcal{D}^e = \{\tau^1, \tau^2, \dots, \tau^m\}, \tau^i = \{s_1^i, s_2^i, \dots, s_n^i\}, s_j^i \in \mathcal{S}$. Denote the transition distribution as $\mu(s, s')$. The objective of LfO can be formulated as a distribution matching problem, finding a policy that minimizes the $f$-divergence between $\mu^\pi(s, s')$ induced by the agent and $\mu^e(s, s')$ induced by the expert [7]:

$$J_{\text{LfO}}(\pi) = \mathbb{E}_{\tau^i \sim \mathcal{D}^e, (s,s') \sim \tau^i} D_f \left[\mu^\pi(s, s') \| \mu^e(s, s')\right]. \tag{3}$$

It is almost impossible to learn a policy directly from the state-only dataset $\mathcal{D}^e$. However, our delicately designed framework (see Figure 1) effectively captures transition features in expert demonstrations to provide informative guidance for RL agents, which will be expounded in the following.

## 3.2 Offline Pretraining Framework

**STG Transformer** is built upon GPT [35] similar to DT [17], but with a smaller scale and more structural modifications to better handle state sequence prediction tasks. Unlike DT, in our setting, neither the action nor the reward can be accessible, so the STG Transformer primarily focuses on predicting the next state embedding given a sequence of states.

As depicted in Figure 2, first we concatenate a few consecutive image frames in the expert dataset to approximate a single state $s_t$. Then, a sequence of $n$ states $\{s_t, \dots, s_{t+n-1}\}$ are encoded into a sequence of $n$ token embeddings $\{e_t, \dots, e_{t+n-1}\}$ by the feature encoder $E_\xi$ composed of several CNN layers and a single-layer MLP, where $e_t = E_\xi(s_t)$. A group of learnable positional embedding parameters is added to the token embedding sequence to remember temporal order. These positional-encoded embeddings are then processed by the causal self-attention module which excels in incorporating information about the previous state sequence to better capture temporal

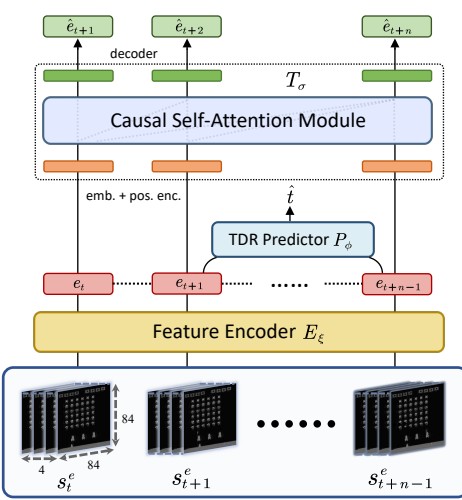

**Figure 2:** State-to-Go Transformer

dependencies, followed by layer normalization. The linear decoder outputs the final latent prediction sequence $\{\hat{e}_{t+1}, \dots, \hat{e}_{t+n}\}$. Denote the positional encoding, transition predicting, and linear decoding model together as $T_\sigma$. It is worth noting that instead of predicting the embeddings of the next state sequence directly, we predict the embedding change and combine it with token embeddings in a residual way, which is commonly applied in transition prediction [4] and trajectory forecasting [36] to improve prediction quality.

For brevity, in further discussion we will refer to $T_\sigma(e_t)$ directly as the predicted $\hat{e}_{t+1}$.

**Expert Transition Discrimination.** Distinguishing expert transiting patterns is the key to leveraging the power of offline expert datasets to improve sample efficiency in downstream RL tasks. Traditional online adversarial methods [3, 12, 13] employ a discriminator to maximize the logarithm probability of transitions sampled from expert datasets while minimizing that from transitions collected online,

which is often sample-inefficient in practice. Moreover, in the case of visual observation, the traditional discriminator may rapidly and strictly differentiate expert transitions from those collected online within a few updates. As a result, the collected observations will be assigned with substantially low scores, making it arduous for policy improvement.

To overcome these limitations, we draw inspiration from WGAN [37] and adopt a more generalized distance metric, known as the Wasserstein distance, to measure the difference between the distributions of expert and online transitions. Compared to the sigmoid probability limited in $[0, 1]$, the Wasserstein distance provides a wider range and more meaningful measure of the difference between two transition distributions, as it captures the underlying structure rather than simply computing the probability. More importantly, unlike traditional online adversarial methods like GAIfO [3] that use the Jensen-Shannon divergence or Kullback-Leibler divergence, the Wasserstein distance is more robust to the issues of vanishing gradients and mode collapse, making offline pretraining possible. Specifically, two temporally adjacent states $s_t, s_{t+1}$ are sampled from the expert dataset, then we have $e_t = E_\xi(s_t), e_{t+1} = E_\xi(s_{t+1})$, and $\hat{e}_{t+1} = T_\sigma(E_\xi(s_t))$. The WGAN discriminator $D_\omega$ aims to maximize the Wasserstein distance between the distribution of expert transition $(e_t, e_{t+1})$ and the distribution of predicted transition $(e_t, \hat{e}_{t+1})$, while the generator tries to minimize it. The objective can be formulated as:

$$\min_{\xi,\sigma} \max_{w \in \mathcal{W}} \mathbb{E}_{\tau^i \sim \mathcal{D}^e, (s_t, s_{t+1}) \sim \tau^i} \left[ D_\omega(E_\xi(s_t), E_\xi(s_{t+1})) - D_\omega(E_\xi(s_t), T_\sigma(E_\xi(s_t))) \right]. \quad (4)$$

$\{D_\omega\}_{\omega \in \mathcal{W}}$ represents a parameterized family of functions that are 1-Lipschitz, limiting the variation of the gradient. We clamp the weights to a fixed box ($\mathcal{W} = [-0.01, 0.01]^l$) after each gradient update to have parameters $w$ lie in a compact space. Besides, to suppress the potential pattern collapse, an additional $L_2$ norm penalizes errors in the predicted transitions, constraining all $e_t$ and $\hat{e}_t$ in a consistent representation space. Thus, the loss functions can be rewritten as follows.

For discriminator:

$$\min_{w \in \mathcal{W}} \mathcal{L}_{dis} = \mathbb{E}_{\tau^i \sim \mathcal{D}^e, (s_t, s_{t+1}) \sim \tau^i} \left[ D_\omega(E_\xi(s_t), T_\sigma(E_\xi(s_t))) - D_\omega(E_\xi(s_t), E_\xi(s_{t+1})) \right]. \quad (5)$$

For STG Transformer (generator):

$$\min_{\xi,\sigma} \mathcal{L}_{adv} + \mathcal{L}_{mse} = - \mathbb{E}_{\tau^i \sim \mathcal{D}^e, s_t \sim \tau^i} D_\omega(E_\xi(s_t), T_\sigma(E_\xi(s_t)))$$
$$+ \mathbb{E}_{\tau^i \sim \mathcal{D}^e, (s_t, s_{t+1}) \sim \tau^i} \| T_\sigma(E_\xi(s_t)) - E_\xi(s_{t+1}) \|_2. \quad (6)$$

By such an approach, the discriminator can distinguish between expert and non-expert transitions without collecting online negative samples, providing an offline way to generate intrinsic rewards for downstream reinforcement learning tasks.

**Temporally-Aligned Representation Learning.** Having a high-quality representation is crucial for latent transition prediction. To ensure the embedding is temporally aligned, we devise a self-supervised auxiliary module, named temporal distance regressor (TDR). Since the time span between any two states $s_i$ and $s_j$ in a state sequence may vary significantly, inspired by [38], we define symlog temporal distance between two embeddings $e_i = E_\xi(s_i)$ and $e_j = E_\xi(s_j)$:

$$t_{ij} = \text{sign}(j - i) \ln(1 + |j - i|). \quad (7)$$

This bi-symmetric logarithmic distance helps scale the value and accurately capture the fine-grained temporal variation. The TDR module $P_\phi$ consists of MLPs with 1D self-attention for symlog prediction. The objective of TDR is to simply minimize the MSE loss:

$$\min_{\xi,\phi} \mathcal{L}_{tdr} = \mathbb{E}_{\tau^i \sim \mathcal{D}^e, (s_i, s_j) \sim \tau^i} \| P_\phi(E_\xi(s_i), E_\xi(s_j)) - t_{ij} \|_2. \quad (8)$$

**Offline Pretraining.** In our offline pretraining, the transition predictor $T_\sigma$ and transition discriminator $D_\omega$ share the same feature encoder $E_\xi$ similar to online methods [39], which allows them to both operate in an easily-predictable and temporally-continuous representation space.

At each training step, a batch of transitions is randomly sampled from the expert dataset. The model is trained autoregressively to predict the next state embedding without accessing any future information. When backpropagating, $\mathcal{L}_{mse}$ and $\mathcal{L}_{adv}$ concurrently update $E_\xi$ and $T_\sigma$ to provide high-quality visual embeddings as well as accurate embedding prediction. $\mathcal{L}_{tdr}$ is responsible for updating the $E_\xi$ and $P_\phi$ as an auxiliary component, and $\mathcal{L}_{dis}$ updates $D_\omega$. Algorithm 1 in Appendix A details the offline pretraining of the STG Transformer.

### 3.3 Online Reinforcement Learning

**Intrinsic Reward.** For downstream RL tasks, our idea is to guide the agent to follow the pretrained STG Transformer to match the expert state transition distribution. Unlike [14], our experimental results show that the pertrained WGAN discriminator is robust enough to clearly distinguish between the sub-optimal online transitions and expert transitions from datasets.

Thus, there is no need for fine-tuning and we just use the discrimination score as the intrinsic reward for online RL. Moreover, we do not use 'progress' like what is done in [9]. This is because, in games with multiple restarts, progress signals can easily be inaccurate and hence mislead policy improvement, while the WGAN discriminator mastering the principle of transitions can often make the correct judgment. The intrinsic reward at timestep $t$ is consequently defined as follows:

$$r_t^i = -\left[ D_\omega\left(E_\xi\left(s_t\right), T_\sigma\left(E_\xi\left(s_t\right)\right)\right) - D_\omega\left(E_\xi\left(s_t\right), E_\xi\left(s_{t+1}\right)\right)\right]. \tag{9}$$

A larger $r_t^i$ means a smaller gap between the current transition and the expert transition.

**Online Learning Procedure.** Given an image observation sequence collected by an agent, the feature encoder first generates corresponding visual representations, followed by the STG Transformer predicting the latent one-step expert transition. Then the discriminator compares the difference between real transitions and predicted transitions. Their Wasserstein distances, as intrinsic rewards $r^i$, is used to calculate generalized advantage, based on which the agent policy $\pi_\theta$ is updated using PPO [40]. It is worth noting that the agent learns the policy merely from intrinsic rewards and environmental rewards are not used.

## 4 Experiments

In this section, we conduct a comprehensive evaluation of our proposed STG on diverse tasks from two environments: a classical **Atari** environment and an open-ended **Minecraft** environment. Among the three mainstream methods mentioned in Section 1, goal-oriented methods are not appropriate for comparison because there is no pre-defined target state. Therefore, we choose IDM-based method **BCO** [19], online adversarial method **GAIfO** [3] and its follow-ups **AMP** [41], **IDDM** [42], and representation-learning method **ELE** [9].

For each task, we conduct 4 runs with different random seeds and report the mean and standard deviation. The same network architecture is applied for each algorithm to maintain consistency. Similar to [39], the discriminator and policy network share the same visual encoder for all online adversarial methods, and we choose one-step prediction for ELE to align with STG. Through extensive experiments, we answer the following questions:

- *Is our proposed framework effective and efficient in visual environments?*
- *Is our offline pretrained discriminator better than the ones which are trained online?*
- *Does TDR make a difference to visual representations? And do we need to add 'progress' rewards, as is done in ELE?*

### 4.1 Atari

**Atari Datasets.** Atari is a well-established benchmark for visual control tasks and also a popular testbed for evaluating the performance of various LfVO algorithms. We conduct experiments on four Atari games: Breakout, Freeway, Qbert, and Space Invaders. To ensure the quality of expert datasets, two approaches are utilized to collect expert observations. For Qbert and SpaceInvaders, we collect the last $10^5$ transitions (around 50 trajectories) from Google Dopamine [43] DQN replay experiences. For Breakout and Freeway, we find the quality of transitions from Dopamine is far from expert. Therefore, we alternatively train a SAC agent [44] from scratch for $5 \times 10^6$ steps and leverage the trained policy to gather approximately 50 trajectories (around $10^5$ transitions) in each game to construct the expert dataset.

**Performance in Atari Games.** As illustrated in Figure 3, STG is on par with BCO in Freeway but outperforms all baselines in the rest tasks. Table 1 displays the final scores of all methods along with expert datasets and PPO learned from environmental rewards. Remarkably, in Breakout

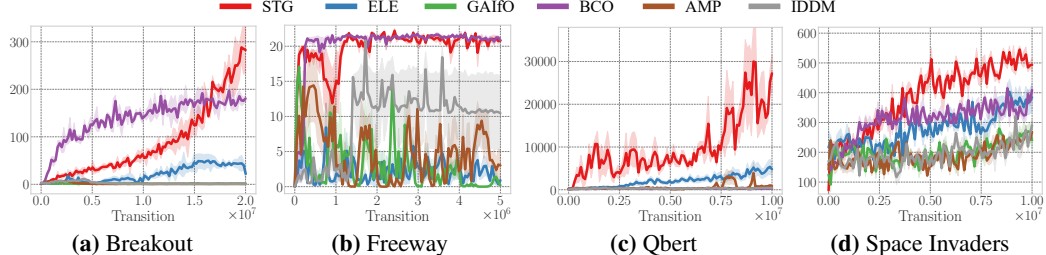

**(a)** Breakout     **(b)** Freeway     **(c)** Qbert     **(d)** Space Invaders

**Figure 3:** STG is compared with BCO, GAIfO, AMP, IDDM, and ELE in four Atari games. The learning curves demonstrate that STG combines the advantage of adversarial learning and the benefit of representation learning, showing substantially better performance in four Atari games.

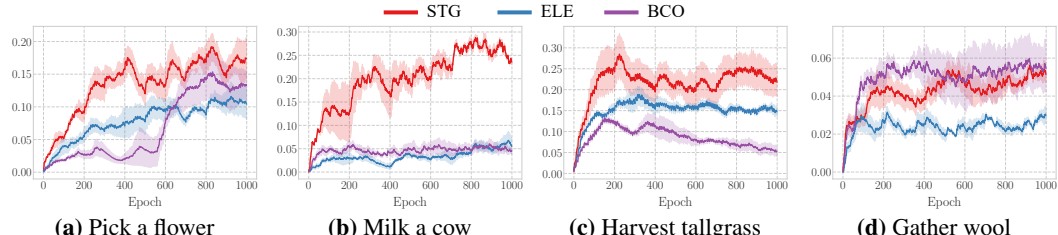

**(a)** Pick a flower     **(b)** Milk a cow     **(c)** Harvest tallgrass     **(d)** Gather wool

**Figure 4:** Average success rates of STG, ELE, and BCO in Minecraft tasks. STG shows superiority over the two baselines in challenging open-ended tasks.

and Qbert, the performance of STG surpasses offline datasets. In Breakout, we observe that STG learns to consecutively bounce the ball up into the top gaps to hit the upper bricks to obtain high intrinsic rewards, while online adversarial methods and ELE fail. In Qbert, STG achieves a prominent breakthrough in the later stages, substantially outperforming other baselines.

**Table 1:** Mean final scores of last 100 episodes in Atari games. The last two columns display the average episodic scores of expert datasets and PPO with environmental rewards reported in [40].

| Environment | GAIfO | AMP | IDDM | ELE | BCO | STG | Expert | PPO |
|---|---|---|---|---|---|---|---|---|
| Breakout | 1.5 | 0.6 | 1.2 | 22.0 | 180.4 | **288.8** | 212.5 | 274.8 |
| Freeway | 0.6 | 3.0 | 10.5 | 2.7 | 21.6 | 21.8 | 31.9 | 32.5 |
| Qbert | 394.4 | 874.9 | 423.3 | 4698.6 | 234.1 | **27234.1** | 15620.7 | 14293.3 |
| Space Invaders | 260.2 | 268.1 | 290.4 | 384.6 | 402.2 | 502.1 | 1093.9 | 942.5 |

**Analysis.** During the training process, we observe that online adversarial methods tend to easily get stuck in a suboptimal policy and struggle to explore a better policy. This is because the discriminator can easily distinguish between the visual behavior of the expert and the imitator based on relatively insignificant factors within just a few online interactions. In contrast, STG learns better temporally-aligned representations in an offline manner, enabling the discriminator to detect more substantial differences. Besides, instead of relying on probability, STG employs the Wasserstein distance metric to provide more nuanced and extensive reward signals. Consequently, even without fine-tuning during the online RL process, STG can offer valuable guidance to the RL agent. Additionally, in Breakout and Freeway we observe that ELE drops in final performance primarily due to the over-optimistic progress, which will be further investigated in Section 4.3. In comparison, STG ensures precise expert transition prediction and discriminative transition judgment, avoiding over-optimistically driving the agent to transfer to new states.

## 4.2 Minecraft

**Brief Introduction.** Built upon Minecraft, Minedojo [45] provides a simulation platform with thousands of diverse open-ended tasks, which propose new challenges for goal completion. Furthermore, the 3D egocentric view and intricate scenes make it extremely hard to extract task-relevant visual signals for LfVO. We evaluate STG on four Minecraft tasks including "pick a flower", "milk a cow",

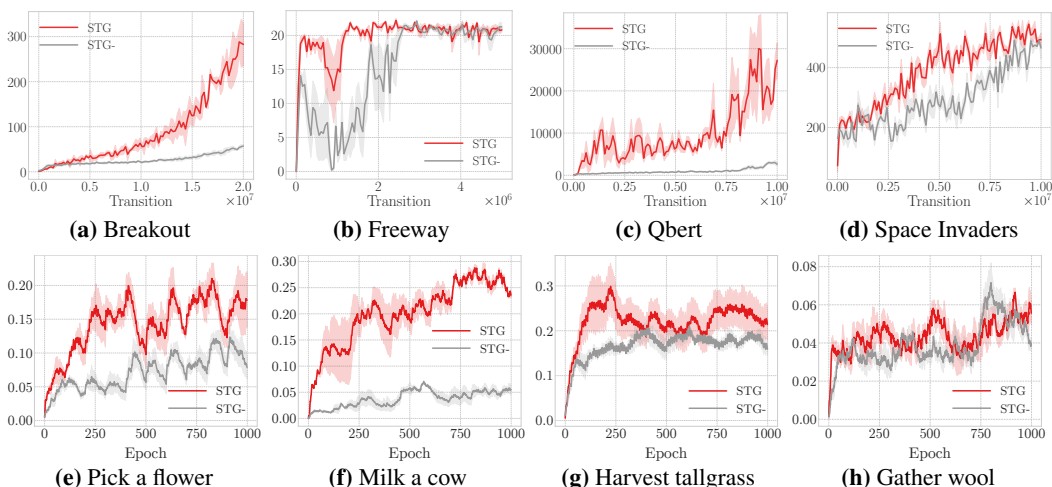

**(a)** Breakout  **(b)** Freeway  **(c)** Qbert  **(d)** Space Invaders

**(e)** Pick a flower  **(f)** Milk a cow  **(g)** Harvest tallgrass  **(h)** Gather wool

**Figure 5:** The removal of the TDR loss from STG, denoted as STG-, induces a decline in performance and sample efficiency in Atari and Minecraft.

"harvest tallgrass", and "gather wool". All tasks are sparse-reward, where only a binary reward is emitted at the end of the episode, thus the performance is measured by success rates.

**Minecraft Datasets.** Recently, various algorithms, *e.g.*, Plan4MC [16] and CLIP4MC [46] have been proposed for Minecraft tasks. To create expert datasets, for each task, we utilize the learned policies of these two algorithms to collect 100 trajectories (around $5 \times 10^4$ observations).

**Performance in Minecraft.** The results on Atari show that online adversarial methods can hardly tackle LfVO tasks well. Therefore, in Minecraft, we focus on comparing BCO, ELE, and STG. As depicted in Figure 4, the success rates across four Minecraft tasks reveal a consistent superiority of STG over ELE. In terms of IDM-based BCO, albeit equipped with an additional online-learned inverse dynamic module, it only shows a negligible advantage over STG in "gather wool" and fails to achieve satisfactory performance in the rest three tasks, and even suffers from a severe performance drop in "harvest tallgras".

Notably, we observe that in "milk a cow", STG attains a success rate approaching 25%, significantly eclipsing the 5% success rate of ELE or BCO. The reasons for this stark contrast are not yet entirely elucidated. However, a plausible conjecture could be attributed to the task's primary objective, *i.e.* locating the cow. STG, which is proficient in memorizing high-quality transitions, can effectively accomplish this subgoal, while ELE may lose the intended viewpoint due to its tendency for over-optimistic progress estimations and BCO can easily get stuck in local online sub-optimal transitions.

### 4.3 Ablations

**TDR Ablation.** Since STG outperforms other baselines in Atari and Minecraft domain, it raises a question regarding the contribution of the designed temporal distance regressor. To answer this question, we conduct an ablation study, denoted as **STG-**, where we remove the TDR loss $\mathcal{L}tdr$ from STG to figure out how the TDR module contributes to enhancing performance and representation. Consequently, the feature encoder $E_\xi$ and the STG Transformer $T_\sigma$ are pretrained under the guidance of a lin-

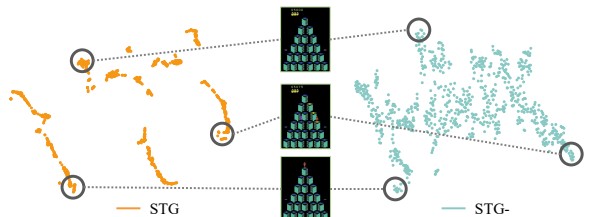

**Figure 6:** T-SNE visualization of a sampled Qbert trajectory embeddings.

ear combination of $\mathcal{L}_{mse}$ and $\mathcal{L}_{adv}$. We keep the other settings fixed and leverage the pretrained model to tackle the respective downstream tasks. The results are shown in Figure 5, where STG is substantially superior to STG- in most tasks.

In order to figure out the underlying reasons for their discrepancy in performance, we compare the visualization of embeddings encoded by STG and STG-. We randomly select an expert trajectory from Qbert and utilize t-SNE projection to visualize their embedding sequences. As illustrated in Figure 6, the embeddings learned by STG exhibit remarkable continuity in adjacent states, significantly different from the scattered and disjoint embeddings produced by STG-. The superior temporal alignment of the STG representation plays a critical role in capturing latent transition patterns, thereby providing instructive information for downstream RL tasks. For more multi-trajectory visualizations, see Appendix G.

**Multi-Task STG.** In the previous experiments, the demonstrations used for pretraining come from agents solving the same tasks, it raises a question on how well do STG generalize to a variety of downstream tasks. To this end, we pretrain a new instance of the STG Transformer named **STG-Multi**, with the same network architecture, on the whole Atari datasets encompassing all four downstream tasks to further assess the efficacy of multi-task adaptation. Considering the four times increase in datasets, we enlarge the model capacity by about four times via stacking more multi-head causal self-attention modules. More details are available in Appendix E. As shown in Figure 7, STG-Multi shows comparable performance across four Atari games. These results strongly reveal the potential of pretraining STG on multi-task datasets for guiding downstream tasks.

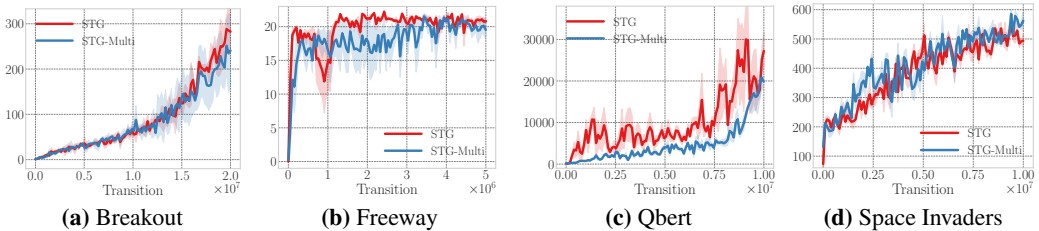

**(a)** Breakout  **(b)** Freeway  **(c)** Qbert  **(d)** Space Invaders

**Figure 7:** Multi-task STG (STG-Multi) is pretrained on the whole Atari datasets to guide RL training in downstream tasks.

**Pretraining Loss Design.** During pre-training, STG concurrently minimizes $\mathcal{L}_{tot} = \alpha \mathcal{L}_{mse} + \beta \mathcal{L}_{adv} + \kappa \mathcal{L}_{tdr}$. We set $\alpha = 0$ to investigate the contribution of $\mathcal{L}_{mse}$, denoted as **STG(rm MSE)**, and $\beta = 0$ to investigate the contribution of WGAN, denoted as **STG(rm Adv)**. For STG(rm Adv), we can only use prediction error as intrinsic rewards like what has been done in [23]. The final performance of STG(rm MSE) and STG(rm Adv) in SpaceInvaders are listed in Figure 8a and 8b. The sample efficiency slightly drops without $L_2$ penalty while the final performance declines heavily without WGAN. Thus, we can conclude each item in $\mathcal{L}_{tot}$ makes an indispensable contribution to the final performance.

It is worth noting that we arbitrarily choose $\alpha = \beta = 0.5$ and $\kappa = 0.1$ in primary experiments according to the value scales of these losses. We discover that these coefficients have a slight influence on training stability and final performance so we do not bother to tune these parameters and leave them consistent across all tasks.

**Datasets.** Given that the reward guidance thoroughly originates from offline dataset, undoubtly the capacity and the quality of the pretraining dataset have a non-negligible impact on performance of LfVO tasks. We first conduct extra ablation studies on reduction of dataset size. 50,000 transitions, i.e. half of the original dataset are randomly sampled to train **STG(Half-Data)** in SpaceInvaders with other settings unchanged. The outcomes of this experiment are presented in Figure 8c. The ablation clearly demonstrates that an increased number of demonstrations contributes to enhanced performance. This observation aligns seamlessly with prior investigations, as illustrated in Figure 4 of [3] and Figure 3 of [19].

Furthermore, we ablate the quality of dataset in Breakout. We manually select 50 trajectories with an episodic return greater than 250 in Breakout to construct expert datasets to train **STG(Expert)** to solve the Breakout game. Figure 8d shows that higher-quality offline datasets contribute to better sample efficiency. Meanwhile, it also reflects that STG can achieve excellent performance learning from sub-optimal observations.

**Progression Reward.** Considering ELE follows the same scenario of pretraining to enhance downstream tasks as STG, we conduct experiments to figure out whether we can additionally add progres-

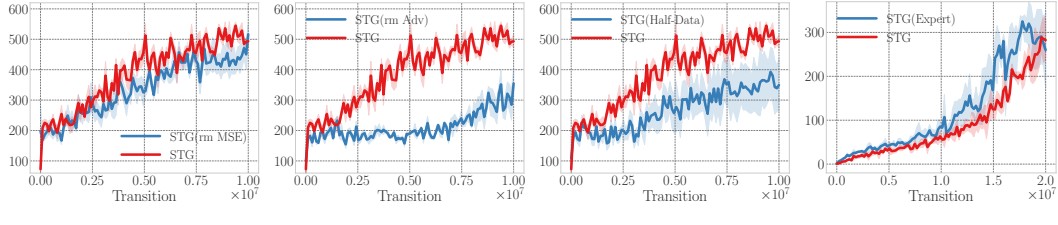

**(a)** Ablate removing $\mathcal{L}_{mse}$ **(b)** Ablate removing $\mathcal{L}_{adv}$ **(c)** Ablate dataset size **(d)** Ablate dataset quality

**Figure 8:** Learning curves of four pre-training ablations: (a) removing $\mathcal{L}_{mse}$ in SpaceInvaders; (b) removing $\mathcal{L}_{adv}$ in SpaceInvaders; (c) using half dataset to train STG in SpaceInvaders; (d) using expert dataset to train STG in Breakout.

sion rewards derived from TDR like ELE to further the performance. To this end, we train the agent under the guidance of both the discriminative and progression rewards from the same pretrained STG Transformer in Atari tasks, denoted as **STG\***. As illustrated in Figure 9, STG outperforms STG\* in all tasks. We analyze that, similar to ELE, progression rewards from TDR over-optimistically urge the agent to "keep moving" to advance task progress, which however can negatively impact policy learning. For example, on certain conditions such as Breakout or Freeway, maintaining a stationary position may facilitate catching the ball or avoiding collision more easily, thereby yielding higher returns in the long run. Therefore, we do not include the over-optimistic progression rewards in our design.

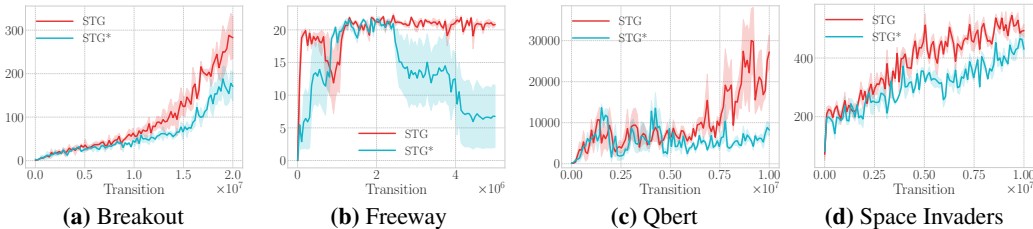

**(a)** Breakout     **(b)** Freeway     **(c)** Qbert     **(d)** Space Invaders

**Figure 9:** Atari experiments comparing using discriminative rewards (STG) and using both discriminative rewards and progression rewards (STG\*).

In summary, our experimental results provide strong evidence for the ability of STG to learn from visual observation, substantially outperforming baselines in a variety of tasks. The ablation study highlights the importance of the TDR module for temporally aligned representations. However, TDR may not be used to generate progression rewards that drive over-optimistic behaviors.

## 5 Conclusion and Future Work

In this paper, we introduce the State-To-Go (STG) Transformer, offline pretrained to predict latent state transitions in an adversarial way, for learning from visual observation to boost downstream reinforcement learning tasks. Our STG, tested across diverse Atari and Minecraft tasks, demonstrates superior robustness, sample efficiency, and performance compared to baseline approaches. We are optimistic that STG offers an effective solution in situations with plentiful video demonstrations, limited environment interactions, and where labeling action is expensive or infeasible.

In future work, STG is likely to benefit from more powerful large-scale vision foundation models to facilitate generalization across a broader range of related tasks. Besides, it can extend to a hierarchical framework where one-step predicted rewards can be employed for training low-level policies and multi-step rewards for the high-level policy, which is expected to improve performance and solve long-horizon tasks.

## Acknowledgments and Disclosure of Funding

This work was supported by NSF China under grant 62250068. The authors would like to thank the anonymous reviewers for their valuable comments.

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

## A  Algorithms

We present our algorithm sketches for STG Transformer offline pretraining and online reinforcement learning with intrinsic rewards respectively.

---

**Algorithm 1** STG Transformer Offline Pretraining

---

**Input:** STG Transformer $T_\sigma$, feature encoder $E_\xi$, discriminator $D_\omega$, expert dataset $D^e = \{\tau^1, \tau^2, \ldots, \tau^m\}, \tau^i = \{s_1^i, s_2^i, \ldots\}$, buffer $\mathcal{B}$, loss weights $\alpha, \beta, \kappa$ .
1: Initialize parametric network $E_\xi, T_\sigma, D_\omega$ randomly.
2: **for** $e \leftarrow 0, 1, 2 \ldots$ **do**                                                              ▷ epoch
3:    Empty buffer $\mathcal{B}$.
4:    **for** $b \leftarrow 0, 1, 2 \ldots |\mathcal{B}|$ **do**                                              ▷ batchsize
5:       Stochastically sample state sequence $\tau^i$ from $D^e$.
6:       Stochastically sample timestep $t$ and $n$ adjacent states $\{s_t^i, \ldots, s_{t+n-1}^i\}$ from $\tau^i$.
7:       Store $\{s_t^i, \ldots, s_{t+n-1}^i\}$ in $\mathcal{B}$.
8:    **end for**
9:    Update $D_\omega$: $\omega \leftarrow \text{clip}(\omega - \epsilon \nabla_\omega \mathcal{L}_{dis}, -0.01, 0.01)$.
10:   Update $E_\xi$ and $T_\sigma$ concurrently by minimizing total loss $\alpha \mathcal{L}_{mse} + \beta \mathcal{L}_{adv} + \kappa \mathcal{L}_{tdr}$.
11: **end for**

---

**Algorithm 2** Online Reinforcement Learning with Intrinsic Rewards

---

**Input:**  pretrained $E_\xi, T_\sigma, D_\omega$, policy $\pi_\theta$, MDP $\mathcal{M}$, intrinsic coefficient $\eta$.
1: Initialize parametric policy $\pi_\theta$ with random $\theta$ randomly and reset $\mathcal{M}$.
2: **while** updating $\pi_\theta$ **do**                                                          ▷ policy improvement
3:    Execute $\pi_\theta$ and store the resulting $n$ state transitions $\{(s, s')\}_t^{t+n}$.
4:    Use $E_\xi$ to obtain $n$ real latent transitions $\{(e, e')\}_t^{t+n}$.
5:    Use $T_\sigma$ to obtain $n$ predicted latent transitions $\{(e, \hat{e}')\}_t^{t+n}$.
6:    Use $D_\omega$ to calculate intrinsic rewards: $\Delta_t^{t+n} = \{D_\omega(e, \hat{e}')\}_t^{t+n} - \{D_\omega(e, e')\}_t^{t+n}$.
7:    Perform PPO update to improve $\pi_\theta$ with respect to $r^i = -\eta \Delta$.
8: **end while**

---

## B  Environment Details

### B.1  Atari

We directly adopt the official default setting for Atari games. Please refer to https://www.gymlibrary.dev/environments/atari for more details.

### B.2  Minecraft

**Environment Settings**

Table 1 outlines how we set up and initialize the environment for each harvest task.

**Table 1:** Environment Setup for Harvest Tasks

| Harvest Item | Initialized Tool | Biome |
|---|---|---|
| milk | empty bucket | plains |
| wool | shears | plains |
| tallgrass | shears | plains |
| sunflower | diamond shovel | sunflower plains |

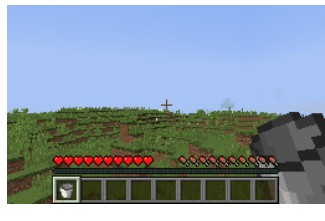
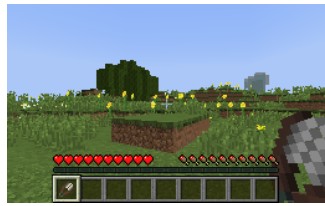

**(a)** Plains            **(b)** Sunflower Plains

**Figure 1:** Biomes in Minecraft

**Biomes.** Our method is tested in two different biomes: plains and sunflower plains. Both the plains and sunflower plains offer a wider field of view. However, resources and targets are situated further away from the agent, which presents unique challenges. Figure 1a and 1b show the biomes of plains and sunflower plains respectively.

**Observation Space.** Despite MineDojo offering an extensive observation space, encompassing RGB frames, equipment, inventory, life statistics, damage sources, and more, we exclusively rely on the RGB information as our observation input.

**Action Space.** In Minecraft, the action space is an 8-dimensional multi-discrete space. Table 2 lists the descriptions of action space in the MineDojo simulation platform. At each step, the agent chooses one movement action (index 0 to 4) and one optional functional action (index 5) with corresponding parameters (index 6 and index 7).

**Table 2:** Action Space of MineDojo Environment

| Index | Descriptions | Num of Actions |
|:---:|:---:|:---:|
| **0** | Forward and backward | 3 |
| **1** | Move left and right | 3 |
| **2** | Jump, sneak, and sprint | 4 |
| **3** | Camera delta pitch | 25 |
| **4** | Camera delta yaw | 25 |
| **5** | Functional actions | 8 |
| **6** | Argument for "craft" | 244 |
| **7** | Argument for "equip", "place", and "destroy" | 36 |

## C Offline Pretraining Details

**Hyperparameters.** Table 3 outlines the hyperparameters for offline pretraining in the first stage.

**Network Structure.** Different architectures for feature encoding are designed for different environments. In Atari, we stack four gray-scale images of shape (84,84) to form a 4-channel state and use the feature encoder architecture as shown in Figure 2a. In Minecraft, a 3-channel image of shape (160,256,3) is directly regarded as a single state, which is processed by a feature encoder with more convolutional layers and residual blocks to capture more complex features in the ever-changing Minecraft world. The detailed structure of the feature encoder for Minecraft is illustrated in Figure 2b. All discriminators, taking in two 512-dimension embeddings from the feature encoder, follow the MLP structure of FC(1024,512)→FC(512,256)→FC(256,128)→FC(128,64)→FC(64,32)→FC(32,1) with spectral normalization.

**Representation Visualization.** We draw inspiration from Grad-CAM [47] to visualize the saliency map of offline-pretrained feature encoder to assess the effectiveness and advantages of the representation of STG. Specifically, we compare the visualization results of STG and ELE in the Atari environment as illustrated in Figure 3. Each figure presents three rows corresponding to the features captured by the three layers of the convolutional layers, respectively. The saliency maps demonstrate that STG exhibits a particular focus more on local entities and dynamic scenarios and effectively ignores extraneous distractions. As a result, compared with ELE, STG shows greater proficiency in

**Table 3:** Hyperparameters for Offline Pretraining

| Hyperparameter | Value |
|---|---|
| STG optimizer | AdamW |
| Discriminator optimizer | RMSprop |
| LR | 1e-4 |
| GPT block size | 128 |
| CSA layer | 3 |
| CSA head | 4 |
| Embedding dimension | 512 |
| Batch size | 16 |
| MSE coefficient | 0.5 |
| Adversarial coefficient | 0.3 |
| TDR coefficient | 0.1 |
| WGAN clip range | [-0.01,0.01] |
| Type of GPUs | A100, or Nvidia RTX 4090 Ti |

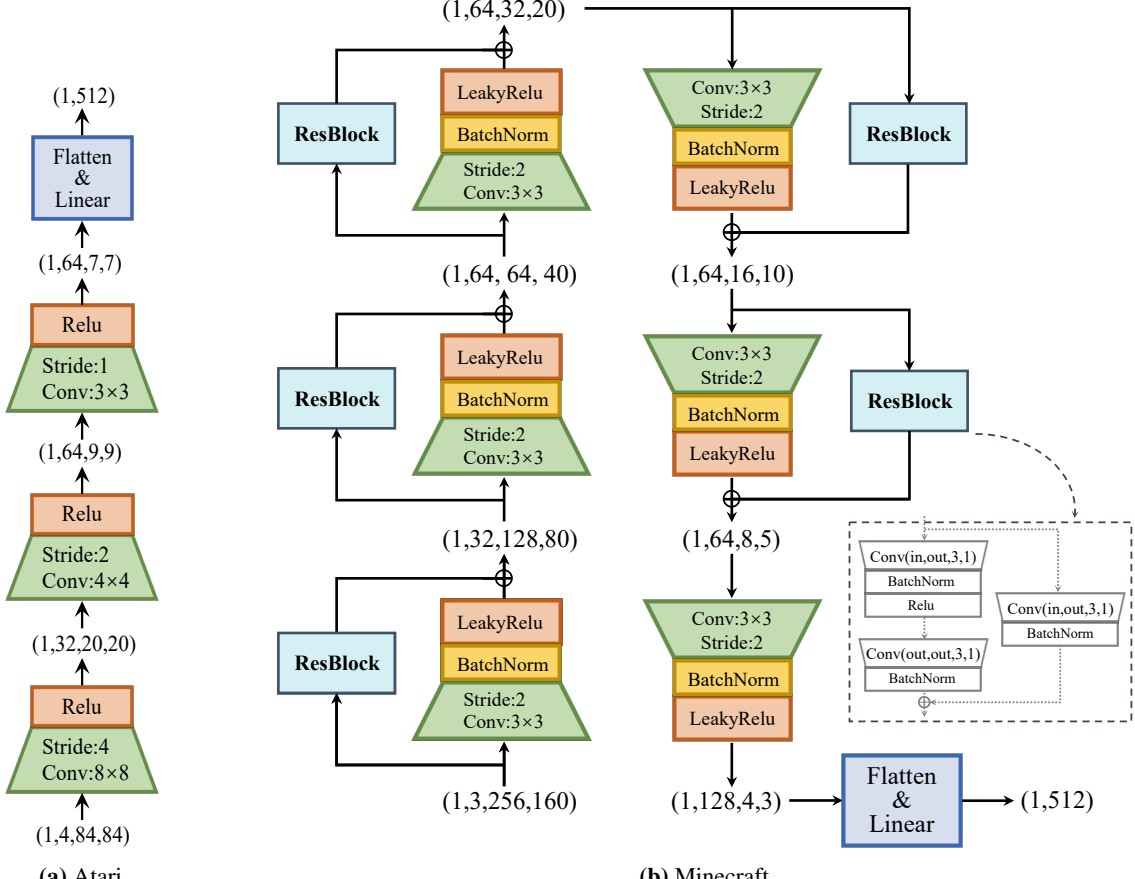

**Figure 2:** Feature encoder structure for Atari and Minecraft

identifying information strongly correlated with state transitions, thereby generating higher-quality rewards for downstream reinforcement learning tasks.

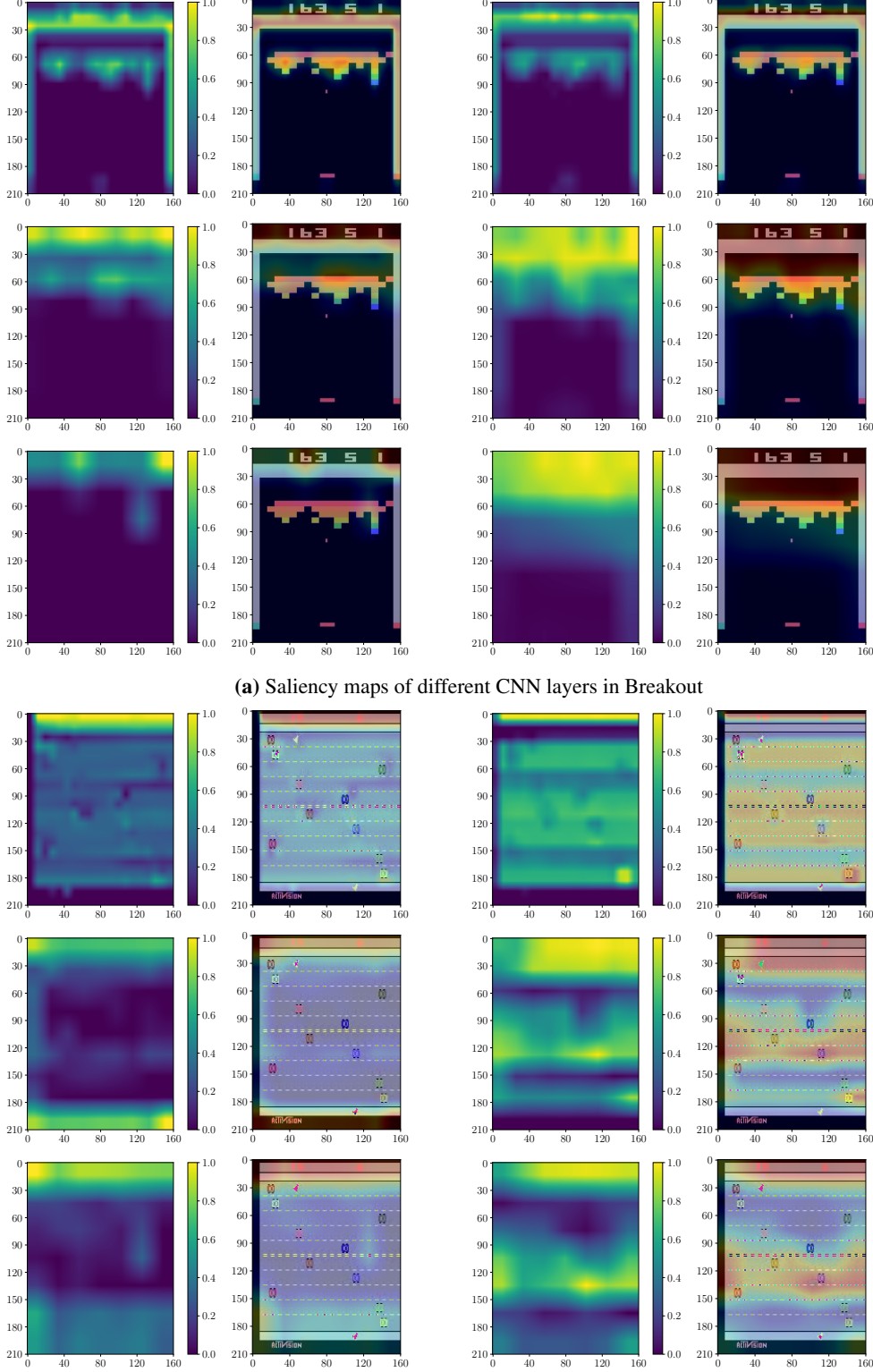

(a) Saliency maps of different CNN layers in Breakout

(b) Saliency maps of different CNN layers in Freeway

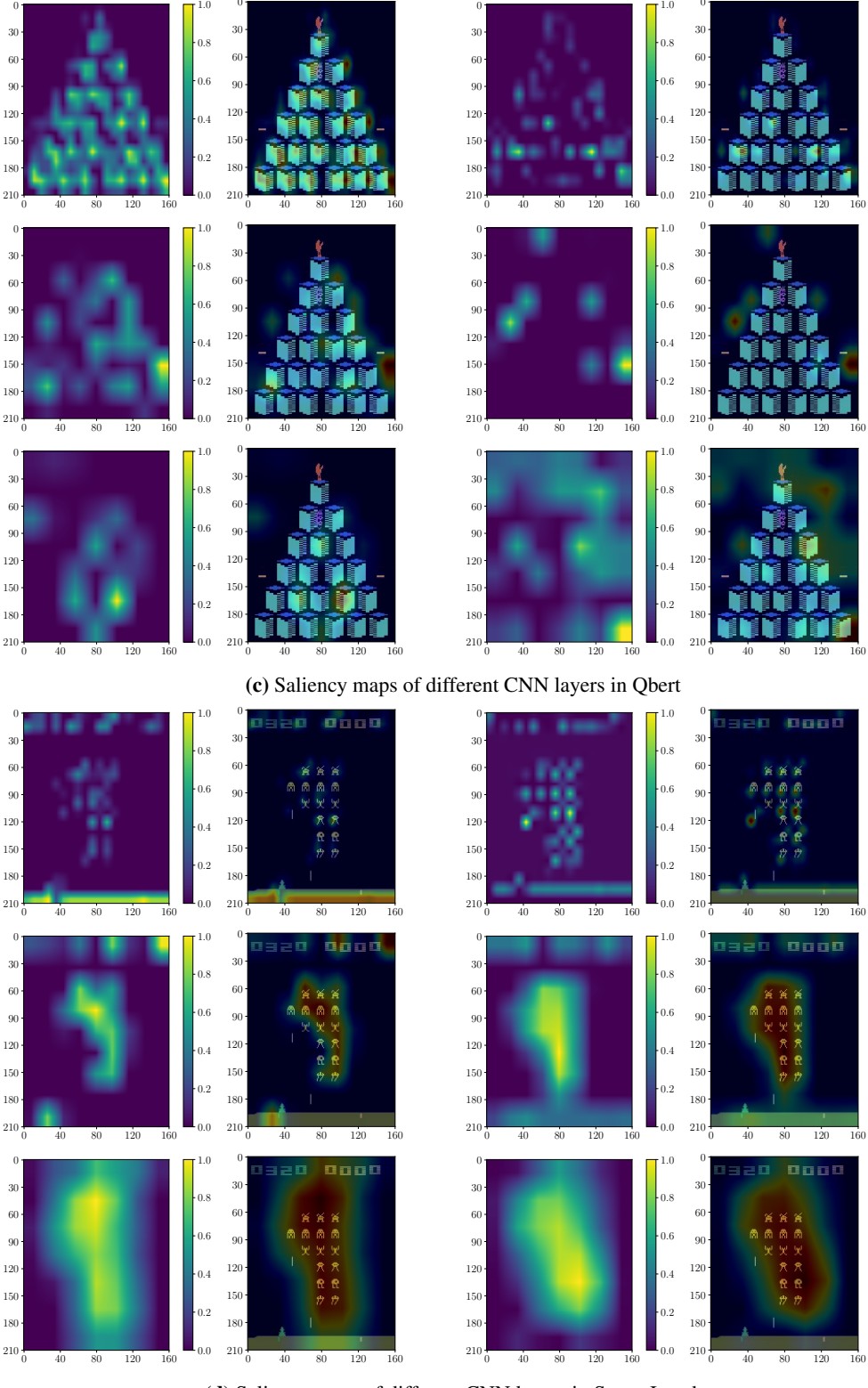

**(c)** Saliency maps of different CNN layers in Qbert

**(d)** Saliency maps of different CNN layers in Space Invaders

**Figure 3:** Saliency maps (SM) of different CNN layers in Atari tasks. The first two columns display the normalized saliency maps and corresponding observations of STG and the last two columns represent SM and corresponding observations of ELE. Through comparison, STG is better at capturing fine-grained features which are strongly correlated with transitions.

# D  RL Training Details

The general training hyperparameters of PPO for downstream RL tasks in the second stage are listed in Table 4.

**Table 4:** General Hyperparameters for PPO

| Hyperparameter | Value |
| --- | --- |
| Optimizer | Adam |
| Learning rate | 2.5e-4 |
| RL discount factor | 0.99 |
| Number of workers (CPU) | 1 |
| Parallel GPUs | 1 |
| Type of GPUs | A100, or Nvidia RTX 4090 Ti |
| Minecraft image shape | (160,256,3) |
| Atari stacked state shape | (84,84,4) |
| Clip ratio | 0.1 |
| PPO update frequency | 0.1 |
| Entropy coefficient | 0.01 |

Neither the discriminative reward from STG nor the progression reward from ELE is bounded. Therefore, it is reasonable to adjust certain hyperparameters to bring out the best performance of each algorithm in each task. In Table 5, the coefficient of intrinsic reward $\eta(\eta > 0)$ for different baselines is tuned to balance the value scale and GAE $\lambda(0 < \lambda < 1)$ is tuned to adjust the impact of intrinsic rewards in different tasks.

**Table 5:** Specific Hyperparameters for Different Tasks

| Task | $\eta_{STG}$ | $\eta_{ELE}$ | $\eta_{GAIfO}$ | $\lambda_{GAE}$ |
| --- | --- | --- | --- | --- |
| Breakout | 0.6 | 1.0 | 2.0 | 0.1 |
| Freeway | 2.0 | 0.1 | 1.0 | 0.15 |
| Qbert | 5.0 | 0.05 | 2.0 | 0.95 |
| Space Invaders | 6.0 | 0.1 | 2.0 | 0.95 |
| Milk a Cow | 1.0 | 0.5 | - | 0.8 |
| Gather Wool | 10.0 | 0.1 | - | 0.8 |
| Harvest Tallgrass | 1.0 | 0.1 | - | 0.95 |
| Pick a Flower | 1.0 | 0.1 | - | 0.95 |

The coefficients of STG* (noted as $\eta r^i + \nu r^*$) in four Atari tasks are reported in Table 6.

**Table 6:** Coefficients for STG* in Atari Tasks

| Task | $\eta$ | $\nu$ |
| --- | --- | --- |
| Breakout | 0.6 | 0.01 |
| Freeway | 2.0 | 0.1 |
| Qbert | 5.0 | 0.03 |
| Space Invaders | 6.0 | 0.01 |

**Training Details.** For Minecraft tasks, we adopt a hybrid approach utilizing both PPO [40] and Self-Imitation Learning (SIL) [48]. Specifically, we store trajectories with high intrinsic rewards in a buffer and alternate between PPO and SIL gradient steps during the training process. This approach allows us to leverage the unique strengths of both methods and achieve superior performance compared to utilizing either method alone [46].

# E   Further Experiments

**Intrinsic Reward Design.** In Equation (9), we define our intrinsic reward $r^i$ as the difference between $r^{guide}$ and $r^{base}$:

$$r^i_t = D_\omega\big(E_\xi(s_t), E_\xi(s_{t+1})\big) - D_\omega\big(E_\xi(s_t), T_\sigma(E_\xi(s_t))\big) = r^{guide}_t - r^{base}_t. \qquad (10)$$

On the one hand, pretrained $D_\omega$ clearly provides informative judgment $r^{guide}$ of transition quality during online interaction. On the other hand, the baseline reward $r^{base}$, solely relying on the current state $s_t$, serves as a baseline to normalize $r^i$ to a relatively lower level. In this section, we aim to investigate the necessity of incorporating $r^{base}$.

To assess the significance of $r^{base}$, we conducted experiments on the four Atari tasks utilizing only $r^{guide}$ as the intrinsic reward, which is similar to previous adversarial methods like GAIfO [3]. In order to bring the scale of $r^{guide}$ in line with $r^i$, we employ running normalization and bound the values within the range of $[-1, 1]$ to mitigate the negative influence of outliers. All other settings remain unchanged. We denote this ablation baseline as STG'.

As illustrated in Figure 4, $r^{guide}$ yields comparable final performance in Breakout and Space Invaders while failing to achieve satisfactory performance in Freeway and Qbert. In contrast, by leveraging $r^{base}$, which provides a unique reference from expert datasets for each individual $s_t$, we observe reduced variance and improved numerical stability compared to the running normalization trick that calculates batch means for normalization.

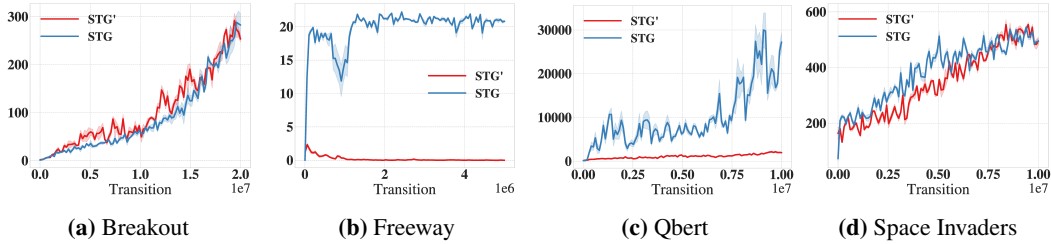

| (a) Breakout | (b) Freeway | (c) Qbert | (d) Space Invaders |

**Figure 4:** Atari experiments comparing using $r^{guide}$ (STG') and $r^i$ (STG) as intrinsic reward.

# F   Additional Ablations

**Multi-Task STG.** Given the four times larger datasets, we enlarge the size of the STG Transformer by increasing the number of heads (24) and layers (16) within the multi-head causal self-attention modules, augmenting the model capacity by about four times. Here we further assess the efficacy of multi-task adaptation of the STG Transformer in Minecraft. We tune the intrinsic coefficient $\eta$ to be 5 for Breakout, Qbert, SpaceInvaders, "milk a cow", "harvest tallgras", "pick a flower", and 10 for Freeway and "gather wool". STG-Multi results in Minecraft are illustrated in Figure 5. As all four tasks share a similar biome in MineDojo, STG-Multi may provide less clear guidance for the agent than STG in downstream tasks, which may cause their discrepancy in performance.

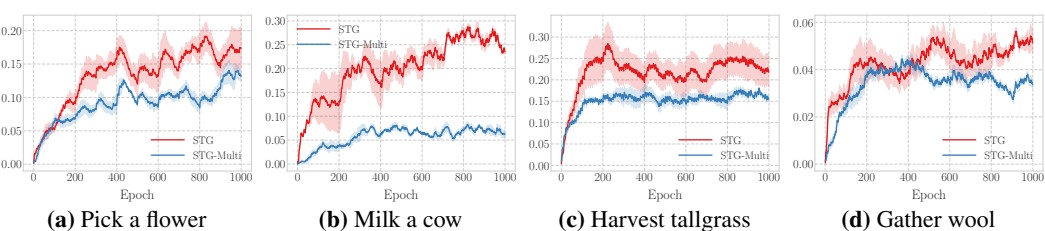

| (a) Pick a flower | (b) Milk a cow | (c) Harvest tallgrass | (d) Gather wool |

**Figure 5:** Multi-task STG (STG-Multi) is pretrained on the whole Minecraft datasets to guide RL training in downstream tasks..

# G  Additional Visualizations

**Intrinsic Return.** As no environmental rewards participate in updating policy, the ultimate objective of LfVO is to maximize the expectation of cumulative dense intrinsic rewards, namely intrinsic return. Figure 6 shows the learning curves of smoothed intrinsic return in Atari and Minecraft. The rising trend proves that online collected observation distribution is getting closer to expert observation distribution during training, indicating the effectiveness of the offline-pretrained Modules.

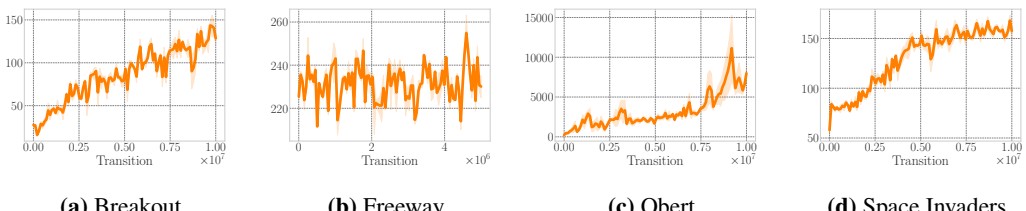

**(a)** Breakout      **(b)** Freeway      **(c)** Qbert      **(d)** Space Invaders

**Figure 6:** The ultimate objective of LfVO is to maximize the expectation of cumulative dense intrinsic rewards, namely intrinsic return. The rising trend proves that online collected observation distribution is getting closer to expert observation distribution.

**Multi-Trajectories Visualization.** As an extension to increase diversity, we additionally visualize more trajectories in SpaceInvaders in Figure 7. We randomly sample five trajectories in the expert dataset and use t-SNE to visualize the embedding sequences encoded by STG and STG- respectively. Expert trajectories of STG exhibit more continuity in adjacent states compared with STG-. This is consistent with the visualization results in Figure 6. Furthermore, it can be observed that after 10M-step of training, the observation distribution is getting closer to expert, and the patterns of STG are closer to exert in comparison with STG-. This reflects that our WGAN-style training indeed generates meaningful reward signal to imitate the behavior of experts and TDR module accelerate the process.

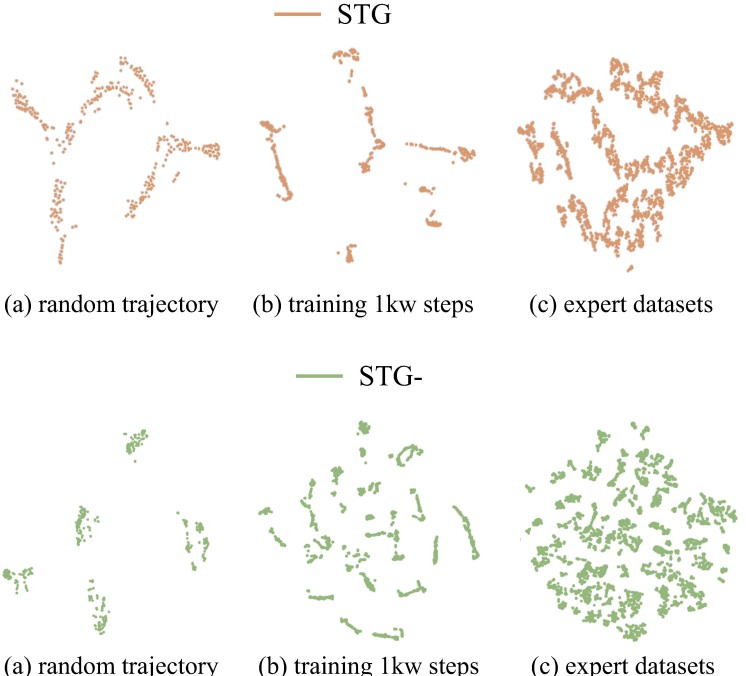

**Figure 7:** T-SNE visualization of embeddings of a random trajectory, a rollout trajectory and expert trajectories in SpaceInvaders.

