# OpenReview forum: "Learning from Visual Observation via Offline Pretrained State-to-Go Transformer"
_NeurIPS.cc/2023/Conference — NeurIPS 2023 poster_

### Official Review · Reviewer_JcJo · 2023-07-02

**Soundness:** 3 good
**Presentation:** 2 fair
**Contribution:** 2 fair
**Rating:** 5
**Confidence:** 4

**Summary:**

This paper proposes a novel approach to deal with the "learning from expert visual observation" setting, lacking action labels. The proposed method first pretrains State-to-Go Transformer (STG), which takes the sequence of encoded visual observations as inputs and predicts next-step embeddings under causal masking. It is trained with adversarial training inspired by WGAN, and with temporally-aligned regressor objectives. The agents are learned, similar to IRL, to follow the action-unlabelled demonstrations. The experiments show the better performance of STG approach on 4 Atari tasks and 4 Minecraft tasks.

**Strengths:**

### Originality
- Incorporating sequence modeling with causal transformer and temporally-aligned regressor into learning from expert visual observation imitation learning seems novel and interesting.

### Quality
- Overall, the paper is well-written and easy to follow.

### Clarity
- Figure 1 helps us understand the brief workflow of the proposed method.

### Significance
- The performance of STG in Atari and Minecraft is better compared to the baselines (ELE, GAIfO, PPO, Expert).
- It would be an insightful observation to the community that combining temporally-aligned regressor into sequence modeling works really well (Figure 5).


**Weaknesses:**

### Originality
- Seeing Figure 3,4,5, the results of STG without temporally-aligned regressor is almost the same as ELE. The most important contribution may come from the off-the-shelf module. Maybe ELE + temporally-aligned regressor would be an interesting ablation if applicable.

### Clarity
- "State-to-Go" might be misleading naming, if it is named as an analogy of Return-to-Go in Decision Transformer. The role of State-to-Go Transformer is quite different from Return-to-Go in Decision Transformer. The agent do not leverage any multi-step-extended values or state representations.
- Minecraft experiments use 50000 demonstrations, while ones on Atari only use 50 demonstrations. There is no description on such a significantly different data size. Also, including the ablation with dataset size is interesting.
- The authors said, "Dopamine is not the expert demonstration" in Section 4.1, but I guess the generated dataset is also not the expert one, seeing Table 4. It is good to include the ablation on dataset quality.


### Significance
- In Minecraft, MineDojo or CLIP4MC builds open-source large-scale pre-trained models that can predict the surrogate reward to measure the similarity between rollouts and textual descriptions. Those are also useful in LfVO settings considered in this paper. In this current situation, the significance of building a small-scale per-task module for surrogate reward is unclear.

**Questions:**

- Why did the authors use PPO for the online RL phase, instead of other algorithms, such as SAC as the authors used to generate the demonstrations in Atari experiments?
- Are STG transformer, discriminator, and encoder frozen during online RL, or co-optimized with online interactions?

**Limitations:**

I think this paper appropriately mentioned the limitations and broader impacts.

---

> ### Author Rebuttal · Authors · 2023-08-09
>
> We thank the reviewer for careful review and constructive suggestions, and for praise our originality "incorporating sequence modeling and temporally-aligned regressor into LfVO" "novel and insightful". We address the questions and concerns below.
>
> `Q1.`"Maybe ELE + temporally-aligned regressor would be an interesting ablation if applicable."
>
> `A1.` In fact, ELE[1] offline learns a temporally-aligned regressor to provide progression rewards for online RL training. STG employs a temporally-aligned regressor to learn high-quality temporally-aligned state representation for transition-prediction module. It offline leans a WGAN discriminator in an adversarial manner to provide intrinsic rewards for downstream RL tasks.
>
> `Q2.`"The most important contribution may come from the off-the-shelf module?"
>
> `A2.` We acknowledge that the TDR module indeed plays a vital role in learning temporally-aligned representation in our work. Without TDR module, the STG-Transformer may not be able to effectively capture latent transition patterns, potentially leading to low-quality discrimination rewards (as discussed in **Section 4.3**). We sincerely hope this insight will prove to be valuable and impactful to the community.
>
> But the most important contribution we want to emphasize is that **unlike most previous adversarial imitation learning methods learning intrinsic rewards online, we offline pretrain STG to effectively and efficiently provide intrinsic rewards guidance for solving difficult visual RL tasks**. We believe this successful attempt highlights the promising future of making an agent "learn like a human". After watching tutorial videos, it can explore with experiences and get down to complete unfamiliar tasks.
>
> `Q3.`The name of "State-to-Go".
>
> `A3.`"State-to-Go" (STG) is named as an analogy to "Return-to-Go" in Decision Transformer (DT), but is not intended for values or representations of states. We use the name "State-to-Go" for the following reasons:
>
> (1) "STG" directly displays the **model property**. We want to stress that STG-Transformer features on modeling pure sequences consisting of continuously adjacent states, rather than modeling (s,a,R) sequence in DT or others in its follow-ups.
>
> (2) "STG" emphasizes the **problem setting**. "State-to-Go" suggests that we focus on learning from state offline datasets without actions or rewards, which is difficult but practical.
>
> (3) "STG" indicates the **algorithm feature**. No matter in offline-pretraining or online learning stage, the key points lie in distinguishing expert or non-expert states. The name STG helps summarize the idea of our algorithm and helps readers understand and memorize our methods.
>
> To address this concern and improve clarity, we will revise the name in our final version as the reviewer suggests.
>
> `Q4.`"Minecraft experiments use 50000 demonstrations, while ones on Atari only use 50 demonstrations".
>
> `A4.`About dataset size, we collect around 50 trajectories, specifically, around 100000 transitions (cf. **Line 254**) in Atari and around 50000 (cf. **Line 299**) transitions in Minecraft. Two datasets are similar in the number of trajectories. **Figure 2(c)** of the attached PDF shows that dataset size indeed makes a difference to the performance.
>
> `Q5.`"Both Dopamine and generated datasets are not expert. It is good to include the ablation on dataset quality."
>
> `A5.`We compare the learning curve between the algorithm [2] for generating Dopamine datasets and our self-implemented SAC. We find that the final performance of [2] drops in the later stages of training and is less competitive than the final performance of SAC. So for Breakout and Freeway we choose to train a SAC agent as expert to provide higher-quality observation datasets. We thank the reviewer for raising an interesting point to ablating dataset quality. As mentioned in Author Rebuttal, **Figure 2(d)** of the attached PDF shows the results ablating dataset quality.
>
> `Q6.`"The significance of building a small-scale per-task module for surrogate reward."
>
> `A6.`In the paper, the reason for pre-training STG in task-specific dataset to solve the corresponding task is to evaluate its capability of offline extracting instructive visual information solely from small observation datasets. To demonstrate that STG can transfer to multi-task setting, we conduct additional experiments by pretraining STG Transformer on Atari datasets encompassing four tasks. The model pre-trained on these multi-task datasets (STG-Multi) is then used to guide RL training for each specific task. To accommodate four-time larger training dataset, we augment the model capacity by increasing the number of heads (24) and layers (16) in self-attention modules. As depicted in **Figure 3** of the attached PDF, the comparable performance reveals the potential of pretraining STG on large-scale multi-task datasets for guiding downstream tasks. With model size scaling up, we firmly believe that STG will become a potent technique to solve larger-scale LfVO tasks, showcasing its versatility and utility in more complex scenarios.
>
> `Q7.`"Choice of RL algorithms."
>
> `A7.`We would like to clarify that any reinforcement method whose target is to maximize the expectation of cumulative discounted rewards can be utilized to generate demonstrations or to optimize the intrinsic return.
>
> `Q8.`"Are STG transformer, discriminator, and encoder frozen during online RL, or co-optimized with online interactions?"
>
> `A8.` After pre-training, modules including the discriminator, encoder, and STG transformer are all frozen and concurrently work as a "reward function" to provide intrinsic rewards for an online RL agent without fine-tuning.
>
> Thanks again for the review! We will implement the feedback in the next version of this paper.
>
> **Reference**
>
> [1] Bruce Jake, et al. "Learning about progress from experts." ICLR 2022.
>
> [2] Rishabh Agarwal, et al. "An optimistic perspective on offline reinforcement learning". ICML 2020

---

> ### Author Response · Authors · 2023-08-14
>
> Dear Reviewer JcJo:
>
> Thanks again for the time and reviews! Since the final stage of discussion is ending soon, please kindly let us know if our response has addressed your concerns.
>
> Sincerely,
>
> Paper3457 Authors

---

> ### Comment · Reviewer_JcJo · 2023-08-15
>
> Thank you for the detailed response and for conducting additional experiments despite the limited time. Here are my remaining concerns.
>
>
> **> A1**
>
> It is quite unclear to understand that ELE also employs TDR to learn progression rewards, from the current paper. I think ELE should be cited in L198, and the difference the author stated here should be clearly described in the Related Work section and Progression Reward paragraph (L331). Because the current manuscript assumes the readers' prior knowledge so much, sufficient information should be contained for self-inclusiveness.
>
> **> A3**
>
> Return-to-go is considered in several previous literature (https://arxiv.org/pdf/1511.05952.pdf, https://arxiv.org/pdf/1911.01546.pdf) before Decision Transformer, and I am still concerned that "state-to-go" may cause the misunderstanding. What is proposed here is a representation learning method for the discriminator of IRL, and it is different from what return-to-go means. Considering another reviewer (HW2S) raises the same concern, I think this should be revised during the discussion period.
>
> **> A4**
>
> Thank you for the clarification, but I could not find the number of trajectories for Minecraft (50?) the authors used in the paper. This also should be included (if you already have it, any pointers would be helpful). The experimental result looks good.
>
>
> **> A6**
>
> Thank you for the multi-task ablations. It would be good to see the promising signs that the proposed method might be scaled up. On the other hand, prior works I mentioned are open-sourced and showed they can be scaled up across more diverse Minecraft domains. I want to put my stance on emphasizing larger-scale experiments in the same game (i.e. MineCraft) domain more (because if we use those pre-trained models, we could learn the policy from the limited dataset).

---

> > ### Author Response · Authors · 2023-08-15
> >
> > Dear Reviewer JcJo:
> >
> > Thanks for the valuable suggestions. Here we present some explanations point by point.
> >
> > > A1
> >
> >    We greatly appreciate the reviewer's valuable feedback, and we will add more detailed descriptions in Related Work section and Progression Reward paragraph to enhance clarity in the final version.
> >
> > > A3
> >
> >    We thank the reviewers to raise this important point. The name "STG" is planned to be substituted by "OLfVO" which highlights the main contribution of **O**ffline **L**earning **f**rom **V**isual **O**bservations. We will revise the name in the final version of the paper.
> >
> > >A4
> >
> >    Thanks for the suggestion. We will add a detailed description of the dataset size in our final version.
> >
> > > A6
> >
> >    We appreciate your recognition of our efforts in conducting scaling-up experiments in Atari. We regret that we were only able to complete the Minecraft experiments due to time constraints. However, we are fully aware that performing multi-task experiments in Minecraft would provide a more comprehensive demonstration of STG's cross-domain capabilities. We are actively engaged in this. We will promptly update the results once available and release the pre-trained models for future research. Moreover, our code is also publicly available. If you would like to check out the code, please let us know and we will provide an anonymous link.
> >
> >    Besides, please kindly note that MineCLIP and CLIP4MC both need text-video pairs for training, our method only relies on videos. Moreover, these two methods are vision-language models (VLMs), providing rewards by the correlation of task text description and visual target. However, in tasks, like Atari games, text-video pairs may not be available to train such VLMs.
> >
> >
> > Thanks again for your time and efforts in making our paper better. If there are still remaining concerns, we are also very happy to have further discussions.
> >
> >
> > Sincerely,
> >
> > Paper3457 Authors

---

> > > ### Author Response · Authors · 2023-08-20
> > > **Whether there are remaining concerns?**
> > >
> > > Dear Reviewer JcJo,
> > >
> > > Considering the discussion is ending soon, we were wondering whether our responses address your concerns. If there are still remaining concerns, we are very happy to have further discussion. If our responses have addressed your concerns, we hope the reviewer will be willing to raise the score.
> > >
> > > Thanks again for your time and efforts in reviewing and improving our work.
> > >
> > > Sincerely,
> > >
> > > Paper3457 Authors

---

### Official Review · Reviewer_B1XQ · 2023-07-03

**Soundness:** 2 fair
**Presentation:** 3 good
**Contribution:** 3 good
**Rating:** 5
**Confidence:** 4

**Summary:**

This paper studies the problem of learning from visual observation. While most adversarial imitation learning methods learn intrinsic reward online, in order to improve the sample-efficiency of online learning, this work proposes to offline pre-train a state-to-go prediction transformer and a discriminator, both conditioned on current states and operating on next states. Representation learning is also enhanced with a shared encoder and an auxiliary temporal distance regression task. Empirical evaluations are conducted in both Atari and Minecraft domains.

**Strengths:**

1. To my knowledge, the proposed offline pretraining framework for adversarial imitation learning is novel and intuitive.
2. Empirical experiments have been conducted in the challenging Minecraft domain.
3. This paper is well-written and easy to follow.

**Weaknesses:**

In my opinion, the most significant weakness is that empirical evaluations are insufficient to support the superiority of the proposed method since some highly relevant but strong baselines are not compared. (see Question 1 below).

**Questions:**

1. Some highly relevant baselines should be compared:
   - BC from observation (BCO) with an IDM: Although the authors claim this method "often suffers from compounding error". A recent work VPT [1] has successfully adopted behavior cloning in the challenging Minecraft domain. Since this work focuses on online imitation learning, BCO, which learns IDM from online interactions, should be compared.
   - Enhanced GAIfO: This paper introduces an offline pretraining framework and State-to-go transformer into adversarial imitation learning. Additional design choices from this work include WGAN and representation learning, which can also equip naive GAIfO. Note that advanced techniques to stabilize adversarial training have been studied by literature. For example, AMP [2] uses a least-squares GAN to mitigate gradient vanishing, similar to this work. Comparison with enhanced GAIfO baselines can further highlight the innovation of the proposed method.
2. It seems the offline adversarial framework in this paper only handles conditional distribution $p(s_{t+1}|s_t)$ rather than marginal distribution $p(s_t, s_{t+1})$ like GAIL, since STG transformer generates samples conditioned on previous states and discriminator contrasts $(s_t, \hat s_{t+1})$ and $(s_t, s_{t+1})$. This may introduce a distribution shift problem. Since at the beginning of online RL, states encountered by the agents can easily deviate from expert distribution, both generator and discriminator have to deal with out-of-distribution states $s_t$. Can the authors provide any intuition or empirical evidence in terms of this problem?

If the authors solve my questions properly, especially for empirical evaluation, I will be happy to increase my rating.

[1] Baker, B., Akkaya, I., Zhokov, P., Huizinga, J., Tang, J., Ecoffet, A., ... & Clune, J. Video pretraining (vpt): Learning to act by watching unlabeled online videos.

[2] Peng, X. B., Ma, Z., Abbeel, P., Levine, S., & Kanazawa, A. Amp: Adversarial motion priors for stylized physics-based character control.

------

Update: The rebuttal from the authors, including additional experimental results and a detailed explanation of the reward design, has addressed most of my issues. Thus I raised my score from 3 to 5.

**Limitations:**

This work has discussed its limitations and future work in the conclusion. There does not seem to be any negative social impact of this paper that should be discussed.

---

> ### Author Rebuttal · Authors · 2023-08-09
>
> We thank the reviewer for careful review and valuable feedback, and for the positive comments that “offline-pretraining framework for adversarial imitation learning is novel and intuitive”. We address the questions and concerns below.
>
> `Q1.` "More empirical evaluations to support the superiority of the proposed method."
>
> `A1.` We would like to explain the reasons for choosing baselines. The main idea of the paper is to **offline pretrain a State-To-Go model extracting visual transition knowledge to provide intrinsic rewards for downstream online RL tasks**. It is the first attempt to offline learn a reward function for online adversarial imitation learning to our knowledge. Thus, in the paper we focus on comparing with methods whose main job is to recover a reward function from visual observations for model-free RL algorithms. Naturally, the IDM-based method BCO [3] **does not fall into the group**, so we choose GAIfO and ELE. Moreover, both GAIfO and ELE **exhibit better performance compared with BCO in their original papers**.
>
> We thank the reviewer for the reasonable suggestions. As shown in the attached PDF **Figure 1**, we compare STG with additional 3 baselines including IDM-based method **BCO** [1], "advanced-GAIfO" **AMP** [2], and **IDDM** [3] in Atari. Just as the reviewer mentioned, BCO is a strong online IDM-based imitation learning baseline that achieves competitive performance in Breakout, Freeway and SpaceInvaders, and "advanced-GAIfO" methods exhibit some advantages compared with GAIfO to some extent. In terms of STG, the final performance is the most competitive among all baselines, and the sample efficiency prominently outperforms most baselines during training. It is worth noting that the final scores of STG in Breakout and Qbert even **exceed expert performance as listed in Table in the paper** while online imitation learning method BCO fails. All these evidences demonstrate STG's remarkable potential of offline learning from visual observations.
>
>
>
> `Q2.` "The offline adversarial framework in this paper only handles conditional distribution rather than marginal distribution?"
>
> `A2.` GAIfO [4] optimizes target $\min_{\pi \in \Pi} \max\_{D \in(0,1)^{{\mathcal{S}} \times \mathcal{S}}} \mathbb{E}\_\pi\left[\log \left(D\left(s, s^{\prime}\right)\right)\right]+ \mathbb{E}\_{\pi_E}\left[\log \left(1-D\left(s, s^{\prime}\right)\right)\right]$ by online learning a transition discriminator to make $p_\pi(s_t,s_{t+1})$ closer to $p_{\pi_E}(s_t,s_{t+1})$. The optimization exactly works on marginal distribution without distribution shift but it suffers from low sample efficiency. To alleviate this problem, we offline pretrain STG-Transformer to predict $\hat{s}_{t+1}$ given $s_t$.
>
> Considering policy conducted by a learning agent may differ from expert policy, during online learning there would exist some "novel" states out of observation dataset distribution. Nevertheless, our **reward design**, denoted as $r_{t}^{i}=D_{\omega}\big( E_{\xi}\left( s_t \right) ,E_{\xi}\left( s_{t+1} \right) \big)-D_{\omega}\big( E_{\xi}\left( s_t \right) ,T_{\sigma}\left( E_{\xi}\left( s_t \right) \right) \big)$, contributes to mitigating this problem. During pre-training, the discriminator learns to effectively distinguish "non-expert" transitions as well as "fake" transitions. At the beginning of online RL, when facing an OOD $s_t$ at timestep $t$, the discriminator will assign relatively low scores to both low-quality online transition $(s_t,s_{t+1})$ and less-accurately predicted transition $(s_t,\hat{s}\_{t+1})$. Consequently, the intrinsic reward $r_{t}^{i}$ almost gets close to zero, which indicates that the OOD $s_t$ will be attached less importance to during RL optimization.
>
> Empirical evaluations also support our intuitions. According to **Figure 4 in the attached PDF**, we visualize the sum of intrinsic rewards, i.e. intrinsic return, of four Atari games respectively with 4 different random seeds. The increasing trend of intrinsic return demonstrates the effectiveness of alleviating distribution shift and guiding RL training. In Freeway, the rising trend is less significant because the episodic return of STG efficiently reaches a plateau in the early stages of training.
>
> To sum up, no matter from intuition or empirical evidence, STG is capable of solving LfVO problems excellently.
>
> Thanks again for the review! We will implement the feedback in the next version of this paper. Further comments are welcome!
>
> **Reference**
>
> [1] Torabi Faraz, et al. "Behavioral Cloning from Observation". arxiv 2018.
>
> [2] Peng, Xue Bin, et al. "Amp: Adversarial motion priors for stylized physics-based character control." ACM 2021.
>
> [3] Yang, Chao, et al. "Imitation learning from observations by minimizing inverse dynamics disagreement." NeurIPS 2019.
>
> [4] Torabi Faraz, et al. "Generative adversarial imitation from observation." CoRR 2018.

---

> > ### Author Response · Authors · 2023-08-10
> > **More detailed description to baseline implementation**
> >
> > Dear Reviewer B1XQ:
> >
> > Thanks again for the time and reviews. Here we want to add more implementation details.
> > Due to the unavailability of public codes, we implement the algorithms AMP and IDDM ourselves.
> > About the network structure, the IDM in BCO, the discriminator in AMP, IDDM and the mutual information estimator in IDDM follow the same network structure in STG to ensure fair comparision.
> >
> > Please kindly let us know if you have any remaining questions or any further concerns.
> >
> > Sincerely,
> >
> > Paper3457 Authors

---

> > ### Comment · Reviewer_B1XQ · 2023-08-10
> >
> > Thanks for the detailed response.
> >
> > I really appreciate the effort made by the authors to compare with additional baselines and it is nice to see that STG outperforms them. I also recommend including experimental results on Minecraft tasks in a future revision.
> >
> > However, here I respectfully request a further explanation for Q2. Why will the discriminator assign relatively low scores to both low-quality online transition $(s_t, s_{t+1})$ and less accurately predicted transition $(s_t, \hat{s_{t+1}})$ for OOD $s_t$? Since $s_t$ is OOD and the discriminator has not been trained on these data, the output of the discriminator should have high uncertainty and not necessarily be low.

---

> > > ### Author Response · Authors · 2023-08-11
> > > **More detailed response to Q2.**
> > >
> > > Dear Reviewer B1XQ:
> > >
> > > We are encouraged by your acknowledgment of our experiments. Your recommendation is valuable. **We are working on the Minecraft tasks**. We will update the results during the discussion period once the results are available. All new results will be included in the next version of this paper.
> > >
> > > We feel sorry that we did not well express our intuitions in Q2. The high uncertainty makes us observe the discrimination score to be at a relatively low level. Actually, both $D_{\omega}\big( E_{\xi}\left( s_t \right), E_{\xi}\left( s_{t+1} \right) \big)$ and $D_{\omega}\big( E_{\xi}\left( s_t \right), T_{\sigma}\left( E_{\xi}\left( s_t \right) \right)$ in $r_i^t$ exist non-negligible uncertainty to some extent as you pointed out. However, the two terms enjoy a similar structure. In detail, the first parameter of $D_{\omega}$ is determined by OOD $s_t$. In terms of the second parameter,  $E_{\xi}\left( s_{t+1} \right)$ depends on the transition from $s_t$ and $T_{\sigma}( E_{\xi}\left( s_t \right))$ depends on single-step prediction also from $s_t$. Considering this factor, both terms in $r_i^t$ may generate quite similar uncertain errors. Under this circumstance, their difference will be relatively low. This kind of intrinsic-reward design helps a lot in mitigating the effects of the OOD states during RL optimization.
> > >
> > > In order to look into the question, we conducted an extra STG experiment in SpaceInvaders to record the value of $r_i^t$ at each timestep $t$ during online RL. We smooth the data and select the values in specific training stages to describe the trend. The results are displayed in the table below. We can observe that during the first 10% of the training process (within 1M steps) the intrinsic reward rapidly jumps from a relatively low level to a higher stage and keeps above 0.011 in the later training stage and finally reaches a top value of 0.01483. This empirically indicates that the designed intrinsic reward is capable of mitigating the problem of distribution shift at the early stage of RL training and keeps promoting policy improvement.
> > >
> > > |                        training stage                        | 0%      | 10%     | 20%     | 30%     | 40%     | 50%     | 60%     | 70%     | 80%     | 90%     | 100%    |
> > > | :----------------------------------------------------------: | ------- | ------- | ------- | ------- | ------- | ------- | ------- | ------- | ------- | ------- | ------- |
> > > | $r_{t}^{i}=D_{\omega}\big( E_{\xi}\left( s_t \right) ,E_{\xi}\left( s_{t+1} \right) \big)-D_{\omega}\big( E_{\xi}\left( s_t \right) ,T_{\sigma}\left( E_{\xi}\left( s_t \right) \right) \big)$ | 0.00643 | 0.01157 | 0.01171 | 0.01115 | 0.01284 | 0.01451 | 0.01253 | 0.01193 | 0.01239 | 0.01181 | 0.01483 |
> > >
> > >
> > > Thanks again for your time and efforts in making our paper better. If we are able to address your concerns, we hope the reviewer will be willing to raise the score. If there are still any remaining concerns, we are also very happy to have further discussions with the reviewer.
> > >
> > >
> > > Sincerely,
> > >
> > > Paper3457 Authors

---

> > > > ### Comment · Reviewer_B1XQ · 2023-08-11
> > > >
> > > > Thanks for the additional response. The detailed explanation for Q2 makes sense to me.
> > > >
> > > > Given that most of my concerns have been addressed, I decided to raise my score.
> > > >
> > > > I also note that a concurrent work [1] proposes a similar method with STG. I encourage the authors to discuss similarities and differences with it in the final version.
> > > >
> > > > [1] Escontrela, A., Adeniji, A., Yan, W., Jain, A., Peng, X. B., Goldberg, K., ... & Abbeel, P. (2023). Video Prediction Models as Rewards for Reinforcement Learning. arXiv preprint arXiv:2305.14343.

---

> > > > > ### Author Response · Authors · 2023-08-11
> > > > >
> > > > > Dear Reviewer B1XQ:
> > > > >
> > > > > We thank the reviewer for raising the score and bringing up this concurrent work. We will add the discussion about similarities and differences in the final version.
> > > > >
> > > > > Sincerely,
> > > > >
> > > > > Paper3457 Authors

---

> > > ### Author Response · Authors · 2023-08-16
> > > **Minecraft Experiments Updates**
> > >
> > > Dear Reviewer B1XQ:
> > >
> > > The experimental results of average success rate on Minecraft task "Milk a cow" and "Gather wool" are ready and we make some updates.
> > >
> > > "Milk a cow"
> > >
> > > | Training Stage | 0%                  | 20%                  | 40%                  | 60%                  | 80%                  | 100%                 |
> > > | -------------- | ------------------- | -------------------- | -------------------- | -------------------- | -------------------- | -------------------- |
> > > | **STG**        | $0.00$%$\pm$$0.00$% | $12.35$%$\pm$$5.80$% | $17.20$%$\pm$$4.89$% | $20.29$%$\pm$$2.39$% | $27.49$%$\pm$$0.71$% | $24.33$%$\pm$$0.21$% |
> > > | **ELE**        | $0.17$%$\pm$$0.12$% | $2.81$%$\pm$$1.04$%  | $1.10$%$\pm$$0.48$%  | $2.65$%$\pm$$0.55$%  | $4.00$%$\pm$$0.42$%  | $5.77$%$\pm$$1.57$%  |
> > > | **BCO**        | $0.13$%$\pm$$0.09$% | $5.58$%$\pm$$1.09$%  | $4.14$%$\pm$$0.64$%  | $4.89$%$\pm$$1.09$%  | $4.67$%$\pm$$0.92$%  | $4.47$%$\pm$$0.48$%  |
> > >
> > > "Gather wool"
> > >
> > > | Training Stage | 0%                  | 20%                 | 40%                 | 60%                 | 80%                 | 100%                |
> > > | -------------- | ------------------- | ------------------- | ------------------- | ------------------- | ------------------- | ------------------- |
> > > | **STG**        | $0.13$%$\pm$$0.11$% | $4.70$%$\pm$$0.96$% | $3.89$%$\pm$$0.56$% | $4.46$%$\pm$$0.65$% | $4.67$%$\pm$$0.51$% | $5.57$%$\pm$$0.68$% |
> > > | **ELE**        | $0.00$%$\pm$$0.00$% | $3.03$%$\pm$$0.40$% | $2.56$%$\pm$$0.59$% | $2.48$%$\pm$$0.51$% | $1.91$%$\pm$$0.29$% | $3.27$%$\pm$$0.66$% |
> > > | **BCO**        | $0.00$%$\pm$$0.00$% | $5.67$%$\pm$$0.88$% | $5.72$%$\pm$$1.44$% | $4.76$%$\pm$$1.85$% | $5.90$%$\pm$$1.04$% | $5.83$%$\pm$$1.49$% |
> > >
> > > Sincerely,
> > >
> > > Paper3457 Authors

---

> > > > ### Author Response · Authors · 2023-08-17
> > > > **Update BCO Experiments on Four Minecraft Tasks**
> > > >
> > > > Dear Reviewer B1XQ:
> > > >
> > > > The experimental results of the average success rates for the four Minecraft tasks are ready and we make some updates.
> > > >
> > > > - "Milk a cow"
> > > >
> > > > | Training Stage | 0%                  | 20%                  | 40%                  | 60%                  | 80%                  | 100%                 |
> > > > | -------------- | ------------------- | -------------------- | -------------------- | -------------------- | -------------------- | -------------------- |
> > > > | **STG**        | $0.00$%$\pm$$0.00$% | $12.35$%$\pm$$5.80$% | $17.20$%$\pm$$4.89$% | $20.29$%$\pm$$2.39$% | $27.49$%$\pm$$0.71$% | $24.33$%$\pm$$0.21$% |
> > > > | **ELE**        | $0.17$%$\pm$$0.12$% | $2.81$%$\pm$$1.04$%  | $1.10$%$\pm$$0.48$%  | $2.65$%$\pm$$0.55$%  | $4.00$%$\pm$$0.42$%  | $5.77$%$\pm$$1.57$%  |
> > > > | **BCO**        | $0.13$%$\pm$$0.09$% | $5.58$%$\pm$$1.09$%  | $4.14$%$\pm$$0.64$%  | $4.89$%$\pm$$1.09$%  | $4.67$%$\pm$$0.92$%  | $4.47$%$\pm$$0.48$%  |
> > > >
> > > > - "Gather wool"
> > > >
> > > > | Training Stage | 0%                  | 20%                 | 40%                 | 60%                 | 80%                 | 100%                |
> > > > | -------------- | ------------------- | ------------------- | ------------------- | ------------------- | ------------------- | ------------------- |
> > > > | **STG**        | $0.13$%$\pm$$0.11$% | $4.70$%$\pm$$0.96$% | $3.89$%$\pm$$0.56$% | $4.46$%$\pm$$0.65$% | $4.67$%$\pm$$0.51$% | $5.57$%$\pm$$0.68$% |
> > > > | **ELE**        | $0.00$%$\pm$$0.00$% | $3.03$%$\pm$$0.40$% | $2.56$%$\pm$$0.59$% | $2.48$%$\pm$$0.51$% | $1.91$%$\pm$$0.29$% | $3.27$%$\pm$$0.66$% |
> > > > | **BCO**        | $0.00$%$\pm$$0.00$% | $5.67$%$\pm$$0.88$% | $5.72$%$\pm$$1.44$% | $4.76$%$\pm$$1.85$% | $5.90$%$\pm$$1.04$% | $5.83$%$\pm$$1.49$% |
> > > >
> > > > - "Harvest tallgrass"
> > > >
> > > > | Training Stage | 0%                  | 20%                  | 40%                  | 60%                  | 80%                  | 100%                 |
> > > > | -------------- | ------------------- | -------------------- | -------------------- | -------------------- | -------------------- | -------------------- |
> > > > | **STG**        | $0.55$%$\pm$$0.11$% | $25.15$%$\pm$$4.56$% | $19.23$%$\pm$$2.63$% | $19.71$%$\pm$$3.61$% | $24.13$%$\pm$$3.58$% | $22.49$%$\pm$$3.43$% |
> > > > | **ELE**        | $1.07$%$\pm$$0.16$% | $15.28$%$\pm$$1.04$% | $15.42$%$\pm$$1.47$% | $15.98$%$\pm$$0.57$% | $15.54$%$\pm$$0.51$% | $14.80$%$\pm$$1.18$% |
> > > > | **BCO**        | $0.97$%$\pm$$0.11$% | $11.89$%$\pm$$0.11$% | $11.21$%$\pm$$2.52$% | $7.83$%$\pm$$0.76$%  | $6.53$%$\pm$$0.95$%  | $5.23$%$\pm$$1.20$%  |
> > > >
> > > > - "Pick a flower"
> > > >
> > > > | Training Stage | 0%                  | 20%                  | 40%                  | 60%                  | 80%                  | 100%                 |
> > > > | -------------- | ------------------- | -------------------- | -------------------- | -------------------- | -------------------- | -------------------- |
> > > > | **STG**        | $0.50$%$\pm$$0.01$% | $10.20$%$\pm$$1.67$% | $17.14$%$\pm$$1.93$% | $13.03$%$\pm$$2.87$% | $17.58$%$\pm$$2.66$% | $17.85$%$\pm$$3.80$% |
> > > > | **ELE**        | $0.17$%$\pm$$0.12$% | $6.45$%$\pm$$1.27$%  | $7.44$%$\pm$$2.67$%  | $9.47$%$\pm$$1.54$%  | $8.67$%$\pm$$1.39$%  | $10.73$%$\pm$$2.87$% |
> > > > | **BCO**        | $0.47$%$\pm$$0.06$% | $2.70$%$\pm$$0.55$%  | $1.75$%$\pm$$1.10$%  | $10.49$%$\pm$$1.39$% | $15.95$%$\pm$$3.99$% | $13.50$%$\pm$$1.85$% |
> > > >
> > > > Sincerely,
> > > >
> > > > Paper3457 Authors

---

### Official Review · Reviewer_HW2S · 2023-07-04

**Soundness:** 2 fair
**Presentation:** 2 fair
**Contribution:** 2 fair
**Rating:** 4
**Confidence:** 4

**Summary:**

This paper presents a new method, called state-to-go (STG) transformer, for learning from visual observation (LfVO), which aims to learn strong policies given the access to (i) offline datasets without actions and rewards and (ii) following online interaction. The main idea of STG transformer is to learn representations by predicting the temporal diference and learn adversarial discriminator that discriminates the embeddings predicted from the model and expert states within the dataset. Then the model is trained to achieve the target tasks by learning to optimize the intrinsic rewards given by the discriminator, which is basically trained to follow the expert behaviors within the offline dataset.

**Strengths:**

Strengths
- 1. Pre-training the discriminator from offline datasets instead of learning the discriminator using the online samples is a novel and interesting idea, to my best knowledge.
- 2. Consistent improvement over the baselines on Atari and Minecraft domains.
- 3. Main description of the method is clear.




**Weaknesses:**

Weaknesses
- 1. This paper is missing crucial analysis that investigates the importance of learning the discriminator using the offline samples. Experiments on Figure 5 (STG, and STG- that does not use temporal representation learning) actually shows that removing representation learning makes performance be very similar to baselines that do not depend on pre-training in several tasks. This makes it questionable whether the proposed offline pre-training component of learning the discriminator is indeed important for the performance and make be think that pre-training good representation learning from offline datasets might be the most important factor. Including more investigations, e.g., experiments with pre-trained encoders for all the methods, could might support that the proposed discriminator training scheme and the intrinsic reward from this are indeed important.
- 2. If the paper wants to claim the newly proposed visual representation learning is one of main contributions, I think proper comparison to prior generic representation learning methods (contrastive learning, masked autoencoding, time contrastive networks, ..) is crucial to claim such a contribution. It makes sense that the proposed method works well but its effectiveness and novelty are not well verified in current status so to be deserved as one of main contributions of the paper, especially considering that using the time index for representation learning is also investigated in prior work ([Cai et al., 2023], [Yun et al., 2022])
- 3. The paper is misusing the term 'State-to-Go' which is not clear what exactly this means. Decision Transformer uses return-to-go as their inputs because their main idea to learn a return-conditioned policy that enables it to stitch the offline samples. But there is no definition of state-to-go in the paper and it is actually not used as inputs or conditioning the policy in any part of the paper. This makes it very difficult to understand the main message from the paper.
- 4. The paper often positions itself as a paper that addresses the difficulty of learning from visual observations on video game domains, while prior work only considers the robotics domains (e.g., line 97). But what exactly are the main factors that make video game domains is more challenging and which experimental evidence supports this? Isn't this claim actually narrowing down the scope of the paper? Also, this should be clearly stated in the abstract or the title that the paper mainly wants to address the main challenges for training offline agents in video game domains.
- 5. This is a minor point, but the paper assumes some background knowledge of prior methods by passing too fastly over the baselines. For instance, the authors might not be familar with ELE and the exact meaning of progression rewards, but the paper assumes such knowledge and does not give a formal definition of progression reward, thus making it difficult to understand the main message from Figure 7 and corresponding subsection.

[Cai et al., 2023] "Open-world multi-task control through goal-aware representation learning and adaptive horizon prediction." CVPR 2023

[Yun et al., 2022] Yun, Sukmin, et al. "Time is matter: Temporal self-supervision for video transformers." ICML 2022

**Questions:**

- Please address the points described in Weaknesses section.
- Predicted embeddings are used as a 'negative' samples in Equation (5) but it's then used as a 'expert' samples in Equation (9). Is this intended, and if so, how could this be justified?

**Limitations:**

Yes.

---

> ### Author Rebuttal · Authors · 2023-08-09
>
> We thank the reviewer for detailed comments and valuable feedback, and for pointing out the novelty of our pre-training scheme. We provide a point-by-point response below.
>
> `Q1.`"Include more investigations, e.g., experiments with pre-trained encoders for all the methods." and "Investigate the importance of learning the discriminator using the offline samples."
>
> `A1.` We would like to clarify that in fact we have taken this point into careful consideration. GAIfO online updates discriminator without pre-training. ELE just pre-trains a temporally-aligned encoder with an auxiliary temporal distance regression task. STG, however, distinguishes itself from GAIfO by incorporating representation learning tricks into offline pre-training paradigm and distinguishes itself from ELE with WGAN-style adversarial learning framework. These prominent differences make STG achieve outstanding performance in offline learning from visual observations, and the performance gap between STG and ELE highlights the importance of learning a WGAN discriminator. Besides, in **Figure 2(b)** in the attached PDF, the signal of transition discrimination from WGAN framework is much stronger than L2 prediction error, which fully proves the significance of adversarial learning.
>
> `Q2.`"comparison to prior generic representation learning methods."
>
> `A2.` The main contribution of the paper is to offer insight to offline learning from visual observations instead of online learning a discriminator like most previous algorithms. To make it work better, we find temporal representation learning is indispensable for the discriminator to provide instructive rewards. Thus, we focus on comparing with intrinsic-reward based methods, and generic representation learning methods do not fall into the group. In **Figure 1** in the attached PDF, we additionally compared STG with more related online adversarial methods **AMP, IDDM** that provide intrinsic rewards and an IDM-based method **BCO**. The high scores and sample efficiency show that STG can efficiently provide effective intrinsic rewards for downstream RL tasks.
>
> `Q3.`The meaning of "State-to-Go".
>
> `A3.` "State-to-Go" (STG) is named as an analogy to "Return-to-Go" in Decision Transformer (DT), but distinct from direct network inputs or policy conditions in DT. We use the name for the following reasons:
>
> (1) "STG" directly displays the **model property**. We want to stress that STG-Transformer features on modeling pure sequences consisting of continuously adjacent states, rather than modeling (s,a,R) sequence in DT or others in its follow-ups.
>
> (2) "STG" emphasizes the **problem setting**. "State-to-Go" suggests that we focus on learning from state offline datasets without actions or rewards, which is relatively difficult but practical.
>
> (3) "STG" indicates the **algorithm feature**. No matter in offline-pretraining or online learning stage, the key points lie in distinguishing expert or non-expert states. The name STG helps summarize the idea of our algorithm and helps readers understand and memorize our methods.
>
> Upon careful consideration of the reviewer's comments, to address this concern and improve clarity, we will revise the name of our algorithm.
>
> `Q4.`"what are the main factors that make video game domains is more challenging (compared with robotic control tasks)? "
>
> `A4.` Video games are challenging in the following aspects:
>
> 1. Complex Dynamics
> 2. Randomness
> 3. Egocentric Visual Observation
> 4. Inaccessible Rewards & Actions
> 5. Benchmarks
>
> Unlike robotic control tasks with deterministic dynamics and accessible instant feedback, video games like Minecraft typically involve intricate and stochastic dynamics that are hard to model. The reward signals in video games are usually sparse and delayed, and the actions are hard to record in most cases. Besides, observations of robotic control usually come from third-person cameras or even proprioceptive states while video games input egocentric images, which makes LfVO a challenging but practical setting in video game domains. Consequently, video games like Atari and Minecraft have become popular benchmarks to evaluate learning from observations or even multi-modal prior knowledge [1,2,3].
>
> `Q5.`"the paper assumes some background knowledge of prior methods by passing too fastly over the baselines and progress reward is not clear."
>
> `A5.` We fastly pass over the baselines for sort out the main line: RL -> IL -> LfO -> LfVO -> offline LfVO. To our knowledge, in ELE, progression is monotonous from beginning to goal, so the progress reward reflects the temporal distance between any two key states. We employ the auxiliary task to learn temporall-aligned representation for STG.
>
> `Q6.`"why predicted embeddings are used as a 'negative' samples in Equation (5) but are used as a 'expert' samples in Equation (9)?"
>
> `A6.` During pre-training phase, the discriminator should learn to distinguish true expert transitions from fake transitions. Therefore, the expert observations serve as positive samples and generated samples serve as negative ones. During online reinforcement learning, especially in the early stage, online collected transitions $(s_t,s_{t+1})\sim \pi_\theta$ are usually far from expert while $(s_t,\hat{s}\_{t+1})$ generated by offline pretrained STG Transformer can work as expert guidance. Under this circumstance, the intrinsic reward $r_i$ measures the gap between the current non-expert transition to an expected high-quality transition. A smaller gap means larger $r_i$.
>
> Thanks again for the review! We will implement the feedback in the next version of this paper. Further comments are welcome!
>
> **Reference**
>
> [1] Hafner, Danijar, et al. "Mastering diverse domains through world models." arxiv 2023.
>
> [2] Ding, Ziluo, et al. "CLIP4MC: An RL-Friendly Vision-Language Model for Minecraft." arxiv 2023.
>
> [3] Yuan, Haoqi, et al. "Plan4mc: Skill reinforcement learning and planning for open-world minecraft tasks." arxiv 2023.

---

> > ### Comment · Reviewer_HW2S · 2023-08-14
> >
> > Thanks for the response and clarification on several points. I have decided to maintain my score for now, but this could be changed during the internal discussion period with other reviewers. Here are my responses to the rebuttal:
> >
> > - As one of the contributions of this paper is on introducing a new representation learning technique, it's still difficult to find why the comparison with other representation learning techniques it not necessary. For instance, you could evaluate the performance by replacing the proposed representation learning with other representation learning methods.
> > - I disagree with the response that states robotic control tasks have instant feedback and third-person cameras, and it's still difficult to see that video games are more challenging or evaluation on video games has a particular superiority to the results on robotics. One of major challenges in reinforcement learning for robotics is in the inherent sparse-reward nature of robotics as the reward design is notoriously difficult and tedious procedure. Also, using a wrist camera, or eye-in-hand camera, is a very popular and widely-used practice in robotics, which leads to recent researches that try to use egocentric large datasets for pre-training in the context of robotics [Xiao et al., 2022; Nair et al., 2022]. It could be nice if the paper could further clarify how the paper is different from prior work that considers robotics domains or further elaborate more on this point.
> > - Thanks for considering to change the name of the method, because it's still honestly not clear the meaning of state-to-go.
> >
> > ---
> >
> > [Xiao et al., 2022] Xiao, Tete, Ilija Radosavovic, Trevor Darrell, and Jitendra Malik. "Masked visual pre-training for motor control." CoRL 2022
> >
> > [Nair et al., 2022] Nair, Suraj, Aravind Rajeswaran, Vikash Kumar, Chelsea Finn, and Abhinav Gupta. "R3m: A universal visual representation for robot manipulation." arXiv preprint arXiv:2203.12601 (2022). CoRL 2022

---

> > > ### Author Response · Authors · 2023-08-15
> > >
> > >
> > >
> > > Dear Reviewer HW2S:
> > >
> > > 1. We thank the reviewer for raising an interesting point to evaluate the performance by replacing the proposed representation learning module. To this end, we substitute the temporal distance regressor with time contrastive network, a contrastive learning method proposed in [1]. The new experiments, denoted as  **STG(w. TCN)**, follow the same pre-training scheme and also use discrimination score for RL with other settings fixed. We record the mean and standard deviation of the episodic return in Freeway and SpaceInvaders respectively.
> > >
> > >    In Freeway (5M steps training):
> > >
> > >    | Training Stage  |      0%       |      25%       |      50%       |      75%       |      100%      |
> > >    | :-------------: | :-----------: | :------------: | :------------: | :------------: | :------------: |
> > >    |     **STG**     | $0.00\pm0.00$ | $20.70\pm0.62$ | $22.20\pm0.63$ | $21.61\pm0.42$ | $21.94\pm1.35$ |
> > >    | **STG(w. TCN)** | $0.00\pm0.00$ | $8.58\pm9.42$  | $14.87\pm8.88$ | $14.70\pm9.04$ | $14.03\pm8.92$ |
> > >
> > >    In SpaceInvaders (10M steps training):
> > >
> > >    | Training Stage  |        0%         |       25%        |       50%        |       75%        |       100%        |
> > >    | :-------------: | :---------------: | :--------------: | :--------------: | :--------------: | :---------------: |
> > >    |     **STG**     |  $72.50\pm41.31$  | $276.29\pm30.82$ | $434.12\pm59.11$ | $514.14\pm70.35$ | $528.67\pm50.77$  |
> > >    | **STG(w. TCN)** | $218.33\pm104.63$ | $262.42\pm38.89$ | $374.43\pm70.15$ | $442.14\pm66.20$ | $450.64\pm119.48$ |
> > >
> > >    The tables above show that prediction-based TDR outperforms contrastive-based TCN. While TCN coarsely distinguishes representations of co-occurring frames from different viewpoints, STG, which predicts temporal distances between adjacent states, excels at capturing more fine-grained variations in specific game scenes. This likely contributes to STG's superior performance.
> > >
> > >    ​
> > >
> > > 2. Besides, we feel sorry that we did not properly distinguish between video games and robotics. In the paper, we mainly compared video game domains with robot simulation domains, such as MuJoCo (cf. **Line 33,88**) most frequently mentioned in related work. In terms of popular MuJoCo tasks, rewards are always easily calculated according to proprioceptive states and typically only third-person observations are accessible, e.g., DeepMind Control Suite. Compared with MuJoCo, video games present greater challenges because of intricate dynamics, sparse rewards, and a mix of egocentric and third-person visual observations.
> > >
> > >
> > >    In sparse-reward robotic manipulation tasks,  we can not agree more that sparse reward imposes much more challenges than MuJoCo tasks. In this domain, existing work also leverages egocentric datasets, but most of them focus on learning better representations for downstream tasks and leave the reward as it is, e.g., the two papers [2,3] the reviewer cited. Unlike these studies, we offline train a discriminator to provide intrinsic rewards for downstream tasks. Thus, our work is orthogonal to these works and can be combined with these methods to additionally provide intrinsic rewards for downstream tasks.
> > >
> > >
> > >
> > > 3. Thanks for advising us for changing the name of "State-To-Go" of our method. The name of "STG" is planned to be substituted by "OLfVO" which highlights the main contribution of offline learning from visual observations. We will revise the name in the final version of the paper.
> > >
> > >
> > >
> > > Thanks again for your time and efforts in making our paper better. If we are able to address your concerns, we hope the reviewer will be willing to raise the score. If there are still any remaining concerns, we are also very happy to have further discussions.
> > >
> > > Sincerely,
> > >
> > > Paper3457 Authors
> > >
> > > **Reference**
> > >
> > > [1] Sermanet, Pierre, et al. "Time-contrastive networks: Self-supervised learning from video." ICRA 2018.
> > >
> > > [2] Xiao, Tete, Ilija Radosavovic, Trevor Darrell, and Jitendra Malik. "Masked visual pre-training for motor control." CoRL 2022.
> > >
> > > [3] Nair, Suraj, Aravind Rajeswaran, Vikash Kumar, Chelsea Finn, and Abhinav Gupta. "R3m: A universal visual representation for robot manipulation." CoRL 2022.

---

> > > > ### Author Response · Authors · 2023-08-17
> > > >
> > > > Dear Reviewer HW2S:
> > > >
> > > > Thanks again for the reviews. Please kindly let us know if our response has addressed your concerns. If you have further concerns, we are looking forward to your reply!
> > > >
> > > > Sincerely,
> > > >
> > > > Paper3457 Authors

---

> > > > ### Comment · Reviewer_HW2S · 2023-08-21
> > > >
> > > > Thank you for your response. I might adjust my final score in the internal discussion phase but I'll maintain my current rating and here are my comments for the reason.
> > > >
> > > > ---
> > > >
> > > > 1. This result is a good starting point but not enough to be conclusive. For instance, there should be more thorough investigation with a variety of baselines with contrastive learning, generative pre-training, image-based pre-training, and video-based pre-training in more environments, as the paper's claim is in proposing a new representation learning method. Or, the other option is to position the paper that proposes a new method that is compatible with any representation learning method and provides a kind of interesting and thorough investigation with multiple underlying methods. But this option needs a new revision.
> > > >
> > > > ---
> > > >
> > > > 2. For your information, Mujoco is a physics engine, not a domain. Basically all domains with sparse rewards, dense rewards, pixel-based inputs, state-based inputs can all use MuJoCo as a physics engine. This was a minor point that requests the paper to further clarify the difference against the prior method by stating that which unique challenge this paper addresses, but it seems like there should be a more understanding on the domains which prior work worked with. Would you consider MuJoCo-based complex humanoid control as 'simpler' than video games? I would not put a too much weight on this for making decisions but it would be nice if you could further clarify on this point.
> > > >
> > > > ---
> > > >
> > > > 3. Thank you.

---

> > > > > ### Author Response · Authors · 2023-08-21
> > > > >
> > > > > Dear Reviewer HW2S:
> > > > >
> > > > > Thanks for the response. We are glad to clarify your concerns.
> > > > >
> > > > > 1. As in https://openreview.net/forum?id=E58gaxJN1d&noteId=0E4sweQP7v `A2`, we claimed that the paper proposed an offline way to solve LfVO tasks, unlike most previous online methods. **Offline pre-training is our main contribution**. In this paper, we utilize a practical temporal representation learning technique similar to TCN [1], ELE [2] to enhance temporal alignment for better prediction and discrimination. The representation learning module is indeed a necessary component as shown by ablation, but not our main contribution. Our key contribution is an **offline** fashion to learn a reward function by predicting the next state. Thus, our experiments primarily focus on evaluating our *offline pre-trained* approach against *online* ones, like GAlfO [3] and additionally BCO [4], AMP [5] suggested by Reviewer B1XQ and IDDM [6] suggested by Reviewer 9y8A. We honestly agree with the reviewer that our method may be compatible with other representation learning methods, e.g., TCN [1] as shown in our previous response. The investigation of a better representation learning module for our offline approach is orthogonal to our focus in this paper.
> > > > > 2. Thanks a lot for your thoughtful reminder. We admit that learning from visual observations is challenging in all domains with sparse rewards and pixel inputs. We concentrate on the video-game domain in this paper and will extend to robotic control tasks in future work.
> > > > >
> > > > > Thanks again for your time and efforts in reviewing and improving our work. If you have further concerns, we are looking forward to your reply!
> > > > >
> > > > > Sincerely,
> > > > >
> > > > > Paper3457 Authors
> > > > >
> > > > > **Reference**
> > > > >
> > > > > [1] Sermanet, Pierre, et al. "Time-contrastive networks: Self-supervised learning from video." ICRA 2018.
> > > > >
> > > > > [2] Bruce Jake, et al. "Learning about progress from experts." ICLR 2022.
> > > > >
> > > > > [3] Torabi, Faraz, et al. "Generative adversarial imitation from observation." CoRR 2018.
> > > > >
> > > > > [4] Torabi, Faraz, et al. "Behavioral Cloning from Observation". arXiv 2018.
> > > > >
> > > > > [5] Peng, Xue Bin, et al. "Amp: Adversarial motion priors for stylized physics-based character control." ACM 2021.
> > > > >
> > > > > [6] Yang, Chao, et al. "Imitation learning from observations by minimizing inverse dynamics disagreement." NeurIPS 2019.

---

> ### Author Response · Authors · 2023-08-14
>
> Dear Reviewer HW2S:
>
> Thanks again for the time and reviews! Since the final stage of discussion is ending soon, please kindly let us know if our response has addressed your concerns.
>
> Sincerely,
>
> Paper3457 Authors

---

### Official Review · Reviewer_9y8A · 2023-07-11

**Soundness:** 2 fair
**Presentation:** 3 good
**Contribution:** 2 fair
**Rating:** 6
**Confidence:** 3

**Summary:**

Paper presents a novel method for learning from visual observations. Specifically, a state encoder is first trained in an offline fashion with two tasks: 1) predicting the normalized relative distance between two states; 2) predicting the relative difference between two consecutive states. Then the whole agent is trained using a standard WGAN-based IRL from observation loss. Experiments on Atari and Minecraft shows some promises against baselines and ablations.

**Strengths:**

+The paper is overall clear and well-written. The authors did a good job illustrating their main idea with stylish graphics and visualization, which is friendly to non-expert readers.

+The method itself is technically sound with good motivations. Although some design choices are not properly backed by ablations, I think the authors explain the rationales under their losses and model architecture well, mostly relevant to the challenges, ex. distinguishing states.

+On the selected arenas, the method indeed show superior performances against the counterparts and the gap is quite significant, suggesting the effectiveness. The TSNE viz is also quite helpful with better understanding on how the method work.



**Weaknesses:**

Having said those above, I do have some major concerns regarding some claims and mostly experiments. I hope the authors can help clarify them in a rebuttal:

-In sec. 3.3, the text reads “That is, the WGAN discriminator can clearly distinguish between the state sequences collected under the learning policy and the expert state sequences, without fine-tuning”. This could be a very strong argument on the proposed representation learning method and the authors seem to derive this from the comparison to GAIfO (w/o the proposed representation learning). But this can be indirect evidence as there might be other factors contributing to the differences, ex. hyper-parameters, additional models, etc. The authors are suggested to include a trajectory visualization that shows if the collected trajectory indeed becomes more distinguishable to expert demonstrations with the proposed representation learning method.

-Baselines; Although I can understand the authors are focusing on IRL-based LfO approach, but indeed I think it is still necessary to include some baselines based off behavior cloning, ex. BCO ([19] in the original paper) and some follow-ups. It would be exciting to see if the proposed representation learning pretraining could help with these BC-based approach as well. Moreover, some additional IRLfO baseline, ex. IDDM ([2] below) and some more recent work are also worth comparing.

-Ablations; The authors only include an ablation that removes the TDR loss from the complete pretraining objective. However, there are many more components and design choices that should be justified by ablations. To name a few:
- transition discrimination losses, L_{mse} and L_{adv}
- coefficient of these losses

-Is It possible to use multiple trajectories for the TSNE visualization in Fig. 6?

-Also Fig. 6, I am a bit confused why the state can be clustered with TDR loss. I will expect a task of predicting the distance between the current state and the goal state (ex. as in [1]) to encourage the emergence of clustering based off the distance-to-go. However, the TDR task is about pair-wise distance between two arbitrary states. Can the authors explain why such loss could facilitate the clusters we observed in Fig. 6?

-citations; some relevant papers on learning Minecraft controllers and LfO in general should be cited: [1-2]

[1] https://arxiv.org/abs/2301.10034

[2] https://arxiv.org/abs/1910.04417


**Questions:**

See [Weaknesses]

**Limitations:**

I am wondering whether the proposed representations losses have to be tied with the transformer architecture, i.e. will they still work with a RNN-style recurrence?

---

> ### Author Rebuttal · Authors · 2023-08-09
>
> We thank the reviewer for detailed comments and insightful review. We are encouraged that the reviewer finds "the idea technically sounds well-motivated in distinguishing states" and "our rationales are clear". We address the questions and concerns below.
>
> `Q1.`"Include some baselines based off behavior cloning and IRLfO baselines."
>
> `A1.` We have added baselines including IDM-based method **BCO**, IRLfO method **IDDM** and "advanced-GAIfO" **AMP** in Atari. The learning curve of STG outperforms all the five baselines in Breakout, Qbert and Space Invaders. In Freeway, the final performance of STG is comparable to BCO and superior to other baselines.
>
> `Q2.` "loss design and coefficients should be justified by ablations."
>
> `A2.` STG concurrently minimize $\mathcal{L}\_{tot}=\alpha\mathcal{L}\_{mse}+\beta\mathcal{L}\_{adv}+\kappa\mathcal{L}\_{tdr}$ during pre-training.
>
> As shown in **Figure 2(a) and 2(b)** in the attached PDF, lack of $\mathcal{L}\_{adv}$ will lead to obvious degradation in performance while lack of $\mathcal{L}\_{mse}$ will lead to slight degradation. Along with TDR ablation, we can find that $\mathcal{L}\_{mse}$ and $\mathcal{L}\_{tdr}$ synergistically contribute to temporally-aligned representation and high-quality sequence prediction, which helps WGAN learn to provide accurate transition discrimination. This is critical for offline learning from visual observations.
>
> About loss coefficients, according to the value scales of these losses, we arbitrarily choose $\alpha=\beta=0.5$ and $\kappa=0.1$ . We discover that these coefficients have a slight influence on training stability and final performance so we do not bother to tune these parameters and leave them consistent across all tasks.
>
> `Q3.` About visualization: "Include a trajectory visualization that shows if the collected trajectory indeed becomes more distinguishable to expert demonstrations with the proposed representation learning method"; "use multiple trajectories"; "why the visualized embeddings can be clustered with TDR loss?"
>
> `A3.` **Figure 5** in the attached PDF displays more visualizations between STG and STG- without TDR module. To increase diversity, we visualize the trajectories in **SpaceInvaders** rather than Qbert in the paper. We randomly sample **5** trajectories in the expert dataset and use t-SNE to visualize the embedding sequences encoded by STG and STG- respectively. Expert trajectories of STG exhibit more continuity in adjacent states compared with STG-. This is consistent with the visualization results in Figure 6 of the paper. From Figure 5(a) to Figure 5(b) in the attached PDF, after 10M-step-training the observation distribution is getting closer to expert, and the patterns of STG are closer to exert in comparison with STG-. This reflects that our WGAN-style training indeed generates meaningful reward signal to imitate the behavior of experts and TDR module accelerate the process.
>
> As for the reason for clustering, the phenomena both occur in Qbert and SpaceInvaders. It is probably related to Atari environment, because in a single Atari task, the distinctive observation patterns are actually countable. With effective representation learning techniques, near-expert observations are likely to cluster into some correct "behavior patterns" for solving a task. This discovery is consistent with previous work [3]. It also describes the task of reaching a key to a door in Montezuma Revenge can be roughly clustered into four patterns with a time-based representation learning skill easily.
>
> `Q4.` "citing some relevant papers."
>
> `A4.` We thank the reviewer for the valuable suggestions. We will include the references[1,2] in our final version.
>
> `Q5.` "Do the proposed representations losses still work with a RNN-style recurrence?"
>
> `A5.` The design of representation learning is orthogonal to our proposed offline pretraining adversarial method.  It will be expected to achieve satisfactory performance no matter applied to RNN-style recurrence or expressive transformer structure.
>
> Thanks again for the review! We will implement the feedback in the next version of this paper. Further comments are welcome!
>
> **Reference**
>
> [1] Shaofei Cai, et al. "Open-World Multi-Task Control Through Goal-Aware Representation Learning and Adaptive Horizon Prediction." CVPR 2023.
>
> [2] Yang, Chao, et al. "Imitation learning from observations by minimizing inverse dynamics disagreement." NeurIPS 2019.
>
> [3] Aytar, Yusuf, et al. "Playing hard exploration games by watching youtube." NeurIPS 2018.

---

> > ### Author Response · Authors · 2023-08-10
> > **Experiments of adjusting coefficients of pre-training loss**
> >
> > Dear Reviewer 9y8A:
> >
> > Thanks again for the time and reviews. Here we want to add more details about the coefficients of pre-training loss
> > $\mathcal{L}\_{tot}=\alpha\mathcal{L}\_{mse}+\beta\mathcal{L}\_{adv}+\kappa\mathcal{L}\_{tdr}$. We explore different coefficient combinations and recorded the average loss over four Atari tasks after training 300,000 steps. As listed in the table, we discover that these coefficients have a slight influence on training stability and final convergence so we do not bother to tune these parameters. According to the value scale of each item, we arbitrarily choose $\alpha=\beta=0.5$ and $\kappa=0.1$ and leave them consistent across all tasks.
> >
> > | $(\alpha,\beta,\kappa)$ | $\mathcal{L}_{mse}$ | $\mathcal{L}_{adv}$ | $\mathcal{L}_{tdr}$ | $\mathcal{L}_{tot}$ |
> > | :---------------------: | :-----------------: | :-----------------: | :-----------------: | :-----------------: |
> > |      (0.5,0.5,0.1)      |       0.0062        |       0.0108        |       0.1341        |       0.02191       |
> > |      (0.5,0.1,0.1)      |       0.0198        |       0.0352        |       0.2174        |       0.03516       |
> > |      (0.1,0.5,0.1)      |       0.0314        |       0.0262        |       0.1850        |       0.03474       |
> > |     (0.5,0.5,0.05)      |       0.0107        |       0.0298        |       0.4494        |       0.04272       |
> >
> > Please kindly let us know if you have any remaining questions or any further concerns.
> >
> > Sincerely,
> >
> > Paper3457 Authors

---

> ### Author Response · Authors · 2023-08-14
>
> Dear Reviewer 9y8A:
>
> Thanks again for the time and reviews! Since the final stage of discussion is ending soon, please kindly let us know if our response has addressed your concerns.
>
> Sincerely,
>
> Paper3457 Authors

---

> > ### Comment · Reviewer_9y8A · 2023-08-16
> >
> > Thanks for the detailed reply! My concerns have been addressed. Please ensure the additional clarifications, results, citations and necessary discussions will be included in the final version. I am happy to raise my score to 6.

---

> > > ### Author Response · Authors · 2023-08-16
> > >
> > > Dear Reviewer 9y8A:
> > >
> > > We are glad to address the reviewer's concerns and thank the reviewer for raising the score. We will include the clarifications, results, citations and necessary discussions in the final version.
> > >
> > > Sincerely,
> > >
> > > Paper3457 Authors

---

### Author Rebuttal · Authors · 2023-08-09

We sincerely thank the four reviewers for their thoughtful comments! We have completed some supplements according to reviewers' suggestions, and we summarize the major changes as follows:

1. We have added three baselines including IDM-based method **BCO**, "advanced-GAIfO" **AMP** and IRLfO method **IDDM** in Atari domain.
2. We ablate the loss items and offline dataset to investigate the factors for good performance in Atari domain.
3. We have conducted multi-task experiments in Atari domain, proving the efficacy of multi-task adaptation of STG.

We provide more details as follows:

1. **About Baselines**

   The main idea we want to convey is to **offline pretrain a State-To-Go model extracting visual transition knowledge to provide intrinsic rewards for downstream online RL tasks**. To our knowledge, it is the first attempt to offline learn a reward function for online adversarial imitation learning. Therefore, in the paper, we focus on comparing methods whose main job is to recover a reward function from visual observations for model-free RL algorithms. Naturally, IDM-based method BCO [3] **does not fall into the group**, so we select two representatives as baselines:

   (1) LfO baseline GAIfO [1] which recovers rewards from online transitions

   (2) recent work ELE [2] which derives rewards from offline representation learning.

   Moreover, both GAIfO and ELE **exhibit better performance compared with BCO in their original papers**. Of course, we believe more baselines help verify the effectiveness of STG. So three baselines including IDM-based method **BCO** [3], "advanced-GAIfO" **AMP** [4], and IRLfO method **IDDM** [5] are added. The learning curves of four Atari tasks shown in **Figure 1** in the attached PDF demonstrate the competitive performance of STG.

2. **About Ablations**

   Incorporating TDR with WGAN is one of the most important designs within our paper. We have conducted an ablation study on the key TDR module to prove its importance. **The influence of other factors appears comparatively less substantial in comparison to TDR or has been investigated in previous literature** (like ablating datasets in [1] and [3]). Upon reasonable suggestions of the reviewers, we have expanded our ablation exploration to encompass loss design and datasets. We find modifications to pre-training loss or datasets lead to analogous variations across different environments. Therefore, we choose one Atari task for each ablation.

   (1) loss design

   During pre-training, STG concurrently minimize $L_{tot}=\alpha L_{mse}+\beta L_{adv}+\kappa L_{tdr}$. We set $\alpha=0$ to investigate the contribution of $L_{mse}$, denoted as **STG(rm MSE)**, and $\beta=0$ to investigate the contribution of WGAN, denoted as **STG(rm Adv)**. For STG(rm Adv), we can only use prediction error as intrinsic rewards like what has been done in [6]. The final performance of STG(rm MSE) and STG(rm Adv) in SpaceInvaders are listed in **Figure 2(a) and 2(b)** in the attached PDF. The sample efficiency slightly drops without L2 penalty while the final performance declines heavily without WGAN. Thus, we can conclude each item in $L_{tot}$ makes an indispensable contribution to the final performance.

   (2) dataset size and quality

   We conduct an ablation study by reducing the size of offline dataset in SpaceInvaders. We randomly sample 50,000 transitions, i.e. half of the original dataset to train **STG(Half-Data)** to solve with other settings unchanged. The outcomes of this experiment are presented in **Figure 2(c)** of the attached PDF. The ablation clearly demonstrates that an increased number of demonstrations contributes to enhanced performance. This observation aligns seamlessly with prior investigations, as illustrated in **Figure 4 of [1] and Figure 3 of [3]**.

   Furthermore, we ablate dataset quality in Breakout. We manually select 50 trajectories with an episodic return greater than 250 in Breakout to construct expert datasets to train **STG(Expert)** to solve Breakout task. **Figure 2(d)** in the attached PDF shows that higher-quality offline datasets contribute to better sample efficiency. Meanwhile, it also reflects that STG can achieve excellent performance learning from sub-optimal observations.

3. **About model and dataset scale-up for multi-task"**

   In the paper, we focus on evaluating STG's capability of offline extracting instructive visual information solely from small observation datasets. To demonstrate that STG can transfer to the multi-task setting, we conduct additional experiments by pretraining STG Transformer on Atari datasets encompassing four tasks. The model pre-trained on these multi-task datasets (STG-Multi) is then used to guide RL training for each specific task. To accommodate a four-time larger training dataset, we augment the model capacity by increasing the number of heads (24) and layers (16) in self-attention modules. As depicted in **Figure 3** in the attached PDF, the comparable performance reveals the potential of pretraining STG on large-scale multi-task datasets for guiding downstream tasks. With model size scaling up, we firmly believe that STG will become a potent technique to solve larger-scale LfVO tasks, showcasing its versatility and utility in more complex scenarios.

All these experiments will be incorporated into the next vision of this paper.

**Reference**

[1] Torabi Faraz, et al. "Generative adversarial imitation from observation." CoRR 2018.

[2] Bruce Jake, et al. "Learning about progress from experts." ICLR 2022.

[3] Torabi Faraz, et al. "Behavioral Cloning from Observation". arxiv 2018.

[4] Peng Xue Bin, et al. "Amp: Adversarial motion priors for stylized physics-based character control." ACM 2021.

[5] Yang Chao, et al. "Imitation learning from observations by minimizing inverse dynamics disagreement." NeurIPS 2019.

[6] Zhu Deyao, et al. "Guiding online reinforcement learning with action-free offline pretraining." arXiv 2023.

---

> ### Author Response · Authors · 2023-08-14
>
> Dear Reviewers,
>
> Thanks again for all the valuable feedback and comments!
>
> Please kindly let us know if you have any remaining questions or any further concerns. We will be more than happy to address your remaining or new concerns and possibly revise our manuscript during the discussion period. If our responses have addressed your concerns, would you mind considering re-evaluating our work based on the updated information?
>
> Sincerely,
>
> Paper3457 Authors

---

### Decision · Program_Chairs · 2023-09-21

**Decision:**

Accept (poster)

**Comment:**

The paper presents a new method, called state-to-go (STG) transformer, for learning from visual observations. The paper in its current form is borderline; based on the reviewers' comments and the authors' rebuttal, I believe the paper has addressed several concerns but still has some aspects that could be further refined.

Reviewer 9y8A expressed concerns about the technical contribution and the limited scope of the proposed method. The authors have appropriately addressed the concerns regarding baselines and ablations, which has led to a positive adjustment in the reviewer's score. However, the reviewer still questions the novelty of the proposed loss function in the broader context of trajectory representation learning. To address this concern, I recommend that the authors emphasize the unique aspects of their loss function within the context of offline pre-training for visual reinforcement learning tasks. While the focus is primarily on Minecraft, discussing potential extensions or applications to other domains, even if speculative, could broaden the paper's appeal.

Reviewer JcJo raised a concern about the terminology used in the paper, particularly "State-to-Go (STG)," and suggested an alternative term such as "OLfVO" (Offline Learning from Visual Observations). The authors should consider this suggestion, especially if it improves clarity and aligns better with the algorithm's nature. Furthermore, the reviewer questioned the significance of a task-specific model in the presence of large-scale pre-trained models in the Minecraft domain. To address this concern, the authors could provide a more explicit comparison with these existing models, highlighting the advantages of their approach in terms of efficiency, performance, or any other relevant metrics.

Reviewer HW2S expressed lingering doubts about the claimed contribution of the proposed representation learning method. While the authors have provided some response to this concern, the reviewer remains uncertain due to the limited number of baselines and tasks evaluated. To address this, I recommend that the authors explore additional baseline methods and tasks to provide a more comprehensive evaluation of their proposed method's performance. This will help strengthen the evidence for the claimed contribution and establish the method's robustness.

In conclusion, the paper shows promise in addressing the challenges of learning from visual observation for open-ended tasks. The authors have effectively responded to several concerns raised by the reviewers, and for this reason my recommendation is for accept. However, there is room for improvement, and the authors should address the remaining issues in time for a camera ready.